# Effects of 36 hours of sleep deprivation on military-related tasks: Can ammonium inhalants maintain performance?

**Jan Maleček**[1]*, **Dan Omcirk**[1], **Kateřina Skálová**[2,3], **Jan Pádecký**[1], **Martin Tino Janikov**[1], **Michael Obrtel**[1], **Michal Jonáš**[1], **David Kolář**[2], **Vladimír Michalička**[1], **Karel Sýkora**[1], **Michal Vágner**[1], **Lubomír Přívětivý**[1], **Tomáš Větrovský**[1], **Zdeňka Bendová**[2,3], **Vít Třebický**[1], **James J. Tufano**[1]

1 Faculty of Physical Education and Sport, Charles University, Prague, Czech Republic, 2 National Institute of Mental Health, Klecany, Czech Republic, 3 Faculty of Science, Charles University, Prague, Czech Republic

* jmalecek@ftvs.cuni.cz

## Abstract

## Introduction

A lack of sleep can pose a risk during military operations due to the associated decreases in physical and cognitive performance. However, fast-acting ergogenic aids, such as ammonia inhalants (AI), may temporarily mitigate those adverse effects of total sleep deprivation (TSD). Therefore, the present study aimed to investigate the acute effect of AI on cognitive and physical performance throughout 36 hours of TSD in military personnel.

## Methods

Eighteen male military cadets (24.1 ± 3.0 y; 79.3 ± 8.3 kg) performed 5 identical testing sessions during 36 hours of TSD (after 0 [0], 12 [–12], 24 [–24], and 36 [–36] hours of TSD), and after 8 [+8] hours of recovery sleep. During each testing session, the following assessments were conducted: Epworth sleepiness scale (ESS), simple reaction time (SRT), shooting accuracy (SA), rifle disassembling and reassembling (DAS), and countermovement jump height (JH). Heart rate (HR) was continuously monitored during the SA task, and a rating of perceived exertion (RPE) was obtained during the JH task. At each time point, tests were performed twice, either with AI or without AI as control (CON), in a counterbalanced order.

## Results

There was faster SRT (1.6%; p < 0.01) without increasing the number of errors, higher JH (1.5%; p < 0.01), lower RPE (9.4%; p < 0.001), and higher HR (5.0%; p < 0.001) after using AI compared to CON regardless of TSD. However, neither SA nor DAS were affected by AI or TSD (p > 0.05). Independent of AI, the SRT was slower (3.2–9.3%; p < 0.001) in the mornings (-24, +8) than in the evening (-12), JH was higher (3.0–4.7%, p < 0.001) in the evenings (-12, -36) than in the mornings (0, -24, +8), and RPE was higher (20.0–40.1%; p < 0.001) in the sleep-deprived morning (-24) than all other timepoints (0, -12, -36, +8).

**Data Availability Statement:** The associated dataset and JASP outputs for all performed analyses are available at the Open Science

Framework [OSF] repository (URL: https://osf.io/
3rj84/; DOI 10.17605/OSF.IO/3RJ84).

**Funding:** This study was supported by the Charles
University Grant Agency (GAUK 986120), the SVV
research program (SVV 2020-2022-260599), the
Cooperatio Program (SPOB research area) and the
Ministry of Health of the Czech Republic (MH CZ-
DRO; NUDZ, 00023752). The funders had no role
in study design, data collection and analysis,
decision to publish, or preparation of the
manuscript.

**Competing interests:** The authors have declared
that no competing interests exist.

Furthermore, higher ESS (59.5–193.4%; p < 0.001) was reported at -24 and -36 than the
rest of the time points (0, -12, and + 8).

## Conclusion

Although there were detrimental effects of TSD, the usage of AI did not reduce those
adverse effects. However, regardless of TSD, AI did result in a short-term increase in HR,
improved SRT without affecting the number of errors, and improved JH while concurrently
decreasing the RPE. No changes, yet, were observed in SA and DAS. These results sug-
gest that AI could potentially be useful in some military scenarios, regardless of sleep
deprivation.

## Introduction

Sleep is a fundamental yet often undervalued physiological process, indispensable for main-
taining physical and mental well-being [1]. It is an essential component of our circadian
rhythms, which have been found to profoundly influence both cognitive and physical perfor-
mance [2]. However, these rhythms can easily be disrupted due to various factors such as
stress, jetlag, travelling, or night-shift work. Such disruptions can result in cognitive and physi-
cal deficits, poorer mental health, and elevated health risks [3, 4]. Therefore, a deeper insight
into the relationships between disruption to sleep and their combined effect on cognitive and
physical performance is of paramount importance. This knowledge, while vital for all, may be
especially critical for at-risk occupations such as military personnel. It has the potential to sig-
nificantly influence their health and combat readiness [5]. Given this, the optimization of
health, well-being, and overall performance could be particularly crucial for them [6, 7]. The
necessity of maintaining adequate sleep quality and quantity thus becomes evident, underscor-
ing the integral role sleep plays in the well-being and performance of military personnel.

Although healthy adults need an average of 7.5 to 8.5 hours of sleep per day [8, 9], self-
reported data indicates that military personnel in various branches of service obtain less sleep
[10], which may have a detrimental effect on the ability to perform military duties efficiently
[11]. Furthermore, a considerable proportion of military personnel is confronted with scenar-
ios where they are required to carry out tasks incessantly and for up to 24 hours [12]. In situa-
tions of military operational necessities such as overnight duty, prolonged operations, or direct
ground combat, some soldiers may endure a lack of sleep that can surpass 24 hours, a condi-
tion known as total sleep deprivation [13]. Research suggests that total sleep deprivation can
decrease blood flow velocity in the middle cerebral artery [14], a major blood vessel that sup-
plies the brain with oxygenated blood. This decrement has been shown to contribute to various
physical, cognitive and behavioral impairments, such as fatigue and impaired sustained atten-
tion and reaction time [14]. Moreover, extant research has indicated that manual dexterity, a
critical component of tasks like shooting and firearms handling, may be negatively impacted
by total sleep deprivation. Previous investigations have reported that the absence of sleep for a
period from 24 to 72 hours can lead to a reduction in shooting accuracy, ranging from 13% to
37% [15, 16] Additionally, another study [17] has demonstrated that even a single night of
total sleep deprivation may have deleterious effects on manual dexterity and hand-eye coordi-
nation, resulting in a decrease in performance by 32%.

In addition to the detrimental effects on cognitive functioning and perceptual-motor skills,
research has demonstrated that total sleep deprivation also impairs short-term, high-intensity
exercise output. Previous research has demonstrated that total sleep deprivation lasting for 24

hours can lead to a reduction of 2% to 10% in 15-meter sprint speed [18–20] and exhaustive time on 400-meter run by 10% [21]. Moreover, after 36 hours of total sleep deprivation, short-term maximal anaerobic performance has been reported to decrease by 5% [22]. Given that soldiers frequently need to perform intense, short-term movements such as sprinting across a battlefield or traversing obstacles in various terrain conditions, a decline in this type of performance may negatively impact a soldier's survival and effectiveness in combat. Since the potential risks and high-stakes nature of military service, the detrimental effects of impaired sleep on cognitive and physical functioning are of significant concern. As such, ergogenic aids, which can alleviate these negative effects by promoting alertness and augmenting physical performance, are of interest to military [23].

One of the most widely used ergogenic aids consumed by military personnel is caffeine [24]. According to prior research, moderate doses of caffeine (approximately 200–300 milligrams) have been found to sustain cognitive functioning, such as alertness and attention [24, 25] or enhance physical performance [25] during sleep deprivation. However, the use of caffeine cannot serve as a replacement for regular sleep. Excessive caffeine consumption can further disrupt sleep patterns, particularly if consumed within six hours prior to bedtime [26, 27]. Furthermore, multiple reviews and meta-analyses [28] show the potential positive effects of caffeine are likely to manifest about 1 hour after ingestion relative to, which may not be sufficient if immediate assistance is required. Therefore, fast-acting forms of ergogenic aids could be beneficial in such circumstances.

An example of a fast-acting ergogenic aid are ammonia inhalants (AI), which are commonly used as a fast-acting "pre-workout stimulant", whereby users hope for rapid improvements in vigilance and short-term high-intensity physical performance [29]. The putative effect of arousal via AI inhalation is believed to be caused by irritation of the respiratory passages that may subsequently trigger the adrenergic receptors in peripheral tissue, resulting in the release of norepinephrine, causing an increase in cardiac output, respiratory rate [30] and an increase in blood flow velocity in the middle cerebral artery [31].

For example, prior research has documented an immediate increase in heart rate [31, 32] and a concurrent increase in middle cerebral artery blood flow velocity [31], following the inhalation of AI, indicating a transient cerebrovascular vasodilation effect that persisted for 15–30 seconds. Despite the absence of evidence indicating that AI inhalation has an impact on maximal muscular strength or endurance [29], some evidence suggests that AI may enhance alertness [32], perceived performance [32], explosive strength during isometric muscle actions [33], as well as repeated anaerobic power performance when athletes were already fatigued [34, 35]. However, no effects were observed in dynamic "real-world" movements such as jump height [32] or sprint time [29]. In addition to that, it is important to note that while AI are widely prevalent among strength-based athletes, there is currently a significant gap in the literature regarding their specific effects on cognitive and physical performance [29].

Therefore this study aims to examine the effectiveness of AI in countering the effects of total sleep deprivation on cognitive and physical performance tests relevant to military personnel. We predict that soldiers will experience decreased cognitive and physical performance with prolonged total sleep deprivation. However, it is expected that the utilization of AI may mitigate these negative effects.

## Methods

### Participants

Eighteen healthy male military cadets (age = 24.1 ± 3.0 years, height = 181.5 ± 6.3 cm, weight = 79.3 ± 8.3 kg, 4.0 ± 0.9 total years of service, all measurements reported as

mean ± SD) serving at the Military department of Charles University participated in this study. The cadets were selected (Q1, 2021) primarily due to their homogenous and synchronized daily cycle based on mandatory morning lineups and the University program. Eligible participants were required to have passed an annual physical fitness test and medical checkup within the last year, have at least two years of active-duty service experience, report a high level of comfort handling firearms, be non-smokers, and currently not working shiftwork or taking medications known to interfere with sleep, cognitive or physical performance. Additionally, the participants selected for this study were all well-acquainted with total sleep deprivation, having experienced it during their military duties. Prior to the study onset, all participants were fully informed about the experimental design and potential risks associated with participation, and provided written informed consent in accordance with the Declaration of Helsinki.

## Experimental design

Data from this study are part of a broader research project aimed at investigating the effects of different light conditions on cognitive and physiological performance during periods of total sleep deprivation. We used a crossover randomized controlled trial design with within-subject repeated-measures to assess the effects of ~36 hours of total sleep deprivation and acute ammonia inhalation on occupationally relevant military tasks in military personnel.

Participants reported to the Sleep and Chronobiology laboratory (National Institute of Mental Health) on Thursday evening after a standardized dinner at ~18:00 h. They then completed a series of questionnaires addressing psychological and physiological health, which were followed by a general familiarization of the layout of the facility (i.e., location of the bathrooms, testing stations, etc.). During this familiarization, the participants were also familiarized with the specific testing procedures and practiced each of the required tasks.

The actual testing protocol began with a night of uninterrupted sleep from ~22:00 h to ~06:30 h. Participants then underwent 5 identical testing sessions from every ~07:30–09:30 h in the morning and ~19:00–21:00 h in the evening. The first test occurred in the morning after the full night of baseline sleep (0 h) and again after 12 hours (-12 h), 24 hours (-24 h), and 36 hours of total sleep deprivation (-36 h) followed by additional testing session after 8 hours (from 22:30 to 06:30 h) of recovery sleep (+8 h). During total sleep deprivation, participants were not allowed to sleep and were kept awake in a common room by passive means, such as playing board games, watching television and reading books while under constant supervision of the research team. Furthermore, the participants were subjected to a constant ambient room light for the entire duration of total sleep deprivation period.

Participants were administered a standardized sleepiness scale and underwent simple reaction time testing, handgun shooting accuracy protocol, a rifle disassembly and reassembly protocol, and countermovement jump testing at each testing session. Participants performed each individual test twice at each testing period, either with AI (AI) or without AI (CON), in randomized order (Fig 1) and separated by 2 minutes of rest (in order to minimize any potential carryover effects of the AI) [31].

For all AI trials, a capsule containing 0.3 mL of AI (composed of 15% of ammonia and 35% of alcohol) [36] was used according to the manufacturer's instructions (Dynarex Corporation, Orangeburg, NY). When the ammonia fumes were released, researcher immediately held the capsule under the participant's nose to inhale until a voluntary withdrawal reflex was observed [31].

During the entire study protocol, participants received personalized daily food rations consisting of standard 'ready to eat' meals commonly used in the Czech military. One week before

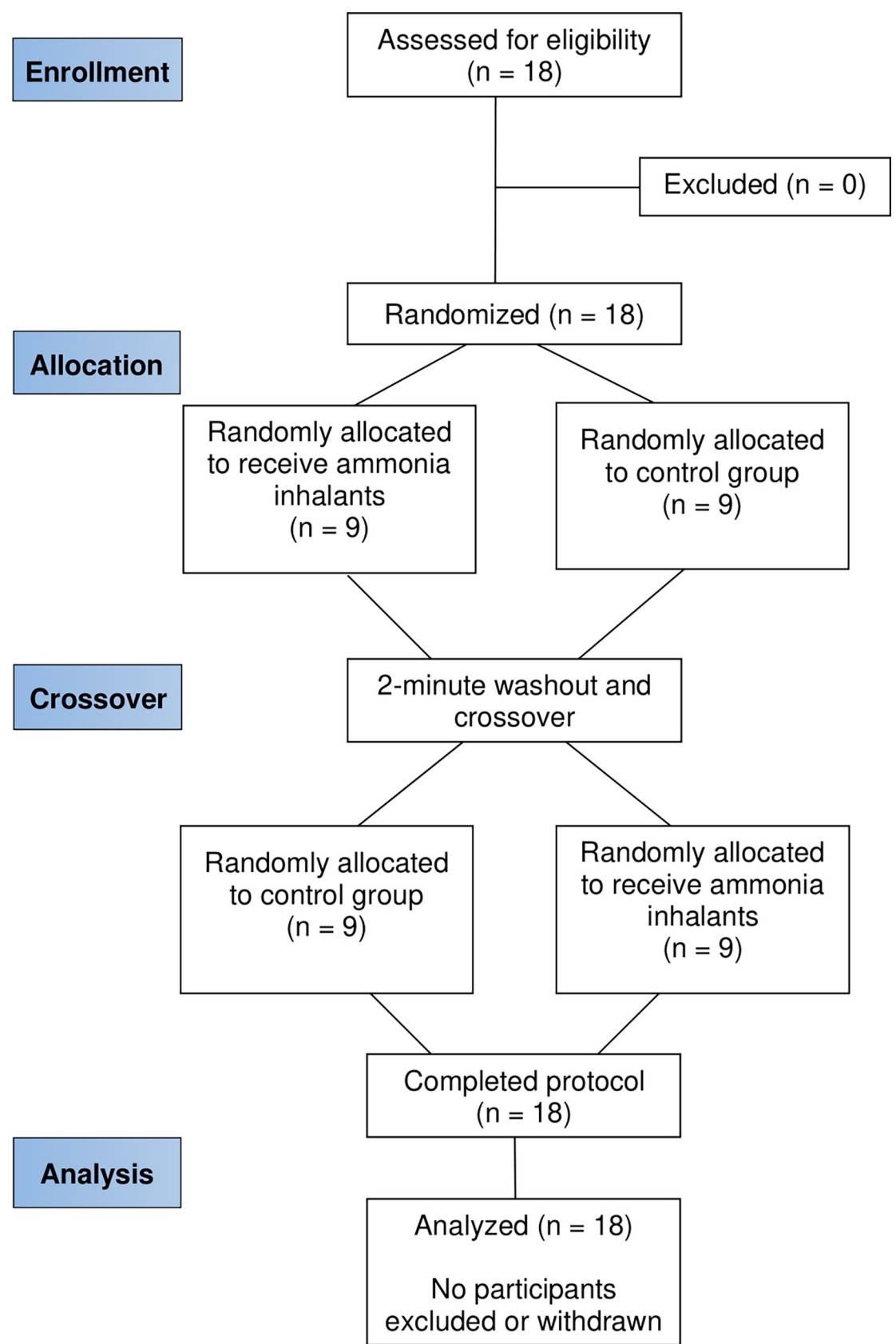

**Fig 1. Flow diagram for crossover trial of each testing period either with ammonia inhalants or without (control group).**

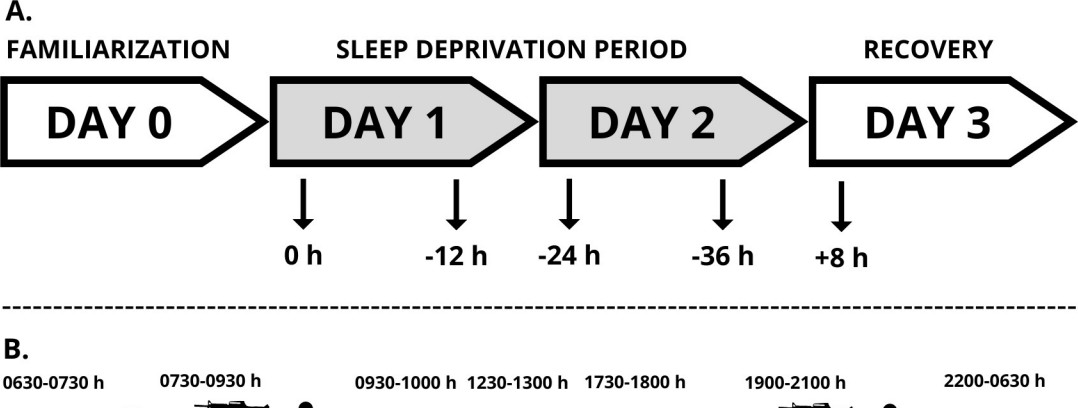

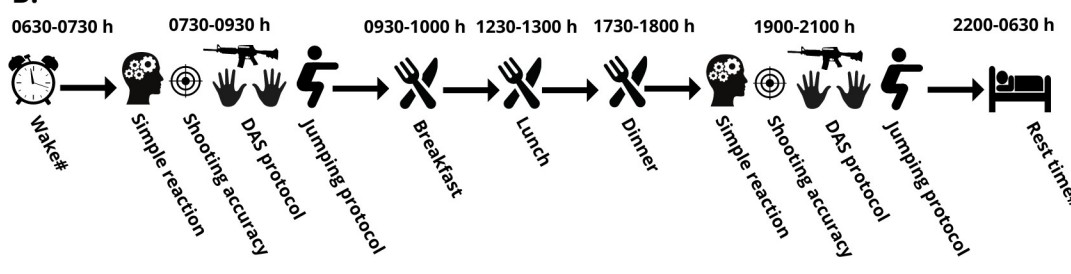

**Fig 2. Overview of the study protocol. (A)** Schedule for days 0–3 of the study, with testing session times (indicated by arrows) at baseline after a full night of sleep (0 h), after 12 (-12 h), 24 (-24 h) and 36 (-36 h) hours of total sleep deprivation followed by 8 (+8 h) hours of recovery sleep. **(B)** Daily timeline of the study. DAS protocol = rifle disassembly and reassembly protocol; #During the total sleep deprivation period between days 1 and 2, participants were not allowed to sleep. Sleep hours on nights leading into day 1 and 3 were from 22:00 h to 06:30 h.

the experiment, participants' body composition was measured (using air displacement plethysmography; Bod Pod Body Composition System; Life Measurement Instruments, Concord, CA), and the total daily energy expenditure was derived from the estimated resting metabolic rate and application of an "active" physical activity factor of 1.6 [37] to the individual caloric requirements. In addition, each participant was allowed ad libitum water consumption. Breakfast was consumed at ~09:30 h, lunch at ~12:30 h, and dinner at ~17:30 h, each day (Fig 2). Additionally, all forms of stimulants were prohibited 72 hours before and during the testing protocol.

## Measurements

**Epworth Sleepiness Scale.** We used Epworth Sleepiness Scale (ESS) translated into Czech. It is a self-administered eight-item questionnaire and takes two to three minutes to complete [38]. The questionnaire presents daily lifestyle activities (i.e. reading, watching TV etc.) and participants rate their current self-perceived likelihood of dozing off in each situation, from: "would never doze" (0) to "high chance of dozing" (3). The ESS provides a cumulative score between 0 and 24, with higher numbers indicating greater sleepiness.

**Simple reaction time.** A simple reaction time test was used to assess the speed of responses to visual stimuli [39]. The evaluation of reaction time was performed using the Psychology Experiment Building Language (PEBL Version 2.0) software [40]. The test consists of instantaneous responses to a visual stimulus by pressing a spacebar key on a laptop's keyboard as quickly as possible when a visual stimulus (white letter X in the middle of the black screen) appears. In the test, 50 trials of stimuli were presented with an interstimulus interval that randomly varied between 250 ms and 2500 ms. Each participant completed four tests (each time two, either with AI or CON, in randomized order) with 2 minutes of inter-test rest. The simple

reaction time data obtained were inspected according to pre-determined criteria, which excluded trial executions that were deemed incorrect due to a reaction time shorter than 150 ms or longer than 3000 ms. The mean reaction time (measured in milliseconds) and the number of incorrect trial executions were used as the variables in the subsequent statistical analysis.

**Shooting protocol.** A laser-based simulator system (LASRX, Plano, TX & Beatrice, NE) with an infrared laser handgun (SIRT 110, Next Level Training, Ferndale, WA) was used (iron sights were used for aiming) to assess handgun shooting accuracy ([41], submitted). Each testing period included two trials of the laser-based handgun shooting protocol (either AI or CON, in randomized order). All trials were performed in the standardized isosceles high-ready stance position (i.e., feet parallel at shoulder-width with both arms extended, holding the handgun with outstretched arms and in front at eye level) [42]. For this study, a real-weight mock-up of the Czech military standard issue Glock 17/22 handgun was used (all participants were familiar with the handgun from their active service). Participants wore over-ear headphones during all testing procedures to hear the software command to start shooting and the simulated shooting blasts when pulling the trigger. A 20 cm circular target was placed on a blank wall 4 meters in front of the participants to simulate a standard-issue 50 cm target 10 meters away for the laser-based handgun shooting protocols (the adjusted size of target was chosen due to limited room dimension available in the research facility). Each trial was separated by 2 minutes (the order of AI and CON were randomized among the participants). For the testing, participants fired 10 shots, aiming to hit the middle of the circular target (a bullseye hit was worth 10 points, and 1 point was deducted for every 1 cm region away from the bullseye, resulting in a maximum score of 100 points). Participants were instructed to try to shoot as accurately as possible within a maximum time limit of 1-minute per trial. The sum of points from each trial was recorded for future analyses.

Additionally, all participants wore a chest strap heart rate monitor (Polar Electro Inc., Model H10, Lake Success, NY, USA) during the shooting protocol. Baseline heart rate data were derived as a mean of heart rate from 2 minutes immediately preceding the start of the shooting trial. Heart rate (bpm) was then continuously monitored during all sessions of shooting protocol. After baseline testing, heart rate data were averaged in 15 seconds bins (0–15, 15–30, 30–45, 45–60) for one minute immediately following the AI and CON trials. The mean of these bins from each trial was used in subsequent analysis [31].

**Jumping protocol.** We used unloaded countermovement jump (CMJ), one of the most common and straightforward strategies to monitor short-term neuromuscular performance in tactical populations [43]. Each CMJ session included 2 sets (AI and CON in a randomized order) of 3 maximal effort CMJs with 2 min of inter-set standing rest. The researcher verbally instructed and encouraged the participants to jump as high as possible on each jump. All CMJs were performed with wooden dowel (~ 0.5 kg) as a mock barbell placed across the participant's upper back mimicking a regular back squat. A linear position transducer (GymAware Power Tool; Kinetic Performance Technologies, Canberra, Australia) was attached to both sides of a dowel to measure the performance. The depth of the CMJ depth was self-selected. Participants wore the same sports t-shirts, shorts and shoes during each test period. The mean of the 3 jump heights (cm) was calculated for each condition at each test session.

In addition, the rating of perceived exertion (RPE) was recorded during the CMJ testing using a CR-10 scale to evaluate RPE scores after each set of CMJ [44]. RPE is a frequently used marker of exercise intensity typically used for monitoring during exercise tests to complement other intensity measures [45].

**Rifle disassembly and reassembly protocol.** The protocol for disassembling and reassembling a military-standard issue assault rifle (specifically the Czech vz. 58 assault rifle) was selected to assess changes in manual dexterity as it is representative tasks that soldiers may

encounter in field operations [46]. During the protocol, participants were tasked to disassemble and reassemble a rifle consisting of 8 parts as fast as possible. Prior to the task's onset, standing participants were instructed to place their hands behind their backs and wait for the researcher's "start" command, after which they attempted to disassemble the rifle as quickly as possible. After a two-minute break, during which participants organized the rifle parts on a table, they then proceeded to reassemble the rifle under the same instruction, and the time was recorded. During the reassembling, the final step was conducting a successful "rifle function check". The time for completion of the task was measured using a handheld stopwatch and recorded on a digital camera for possible corrections.

Each participant completed two trials of rifle disassembly and reassembly (AI and CON, in a randomized order) with two minutes of rest between conditions. The performance measure used in this study was the sum of the disassembly and reassembly time in seconds for each condition (AI and CON).

To assess the reliability of the task, participants familiarize themselves with the disassembly and reassembly protocol before the study begins. They repeatedly performed the protocol for three consecutive days, one week before participating in the study. Their performance showed sufficient reliability (data and reliability analysis of the familiarization period can be found in the supplementary materials).

During the testing protocol, data were recorded for 14 of the 18 participants. The remaining four participants were unable to participate in the familiarization due to service duties and were thus excluded from the respective subsequent analysis.

## Statistical analysis

Statistical analyses were conducted using JASP (version 0.16.2, 2022) [47]. Parametric tests were performed once the normality assumptions were verified using the Shapiro-Wilk test. Data were analyzed using a two-way repeated measures ANOVA (2 conditions: [AI, CON] × 5 time: [0 h, -12 h, -24 h, -36 h, +8 h]). Heart rate values were analyzed using a three-way repeated measures ANOVA (2 conditions: [AI, CON] × 5 time: [0 h, -12 h, -24 h, -36 h, +8 h] × 4 time spans: [0–15, 15–30, 30–45, 45–60 sec]). Lastly, the heart rate percentage difference values were analyzed using a one-way repeated measures ANOVA (1 condition: [heart rate percentage difference between AI and CON] × 4 time spans: [0–15, 15–30, 30–45, 45–60 sec]). Sphericity was assessed using Mauchly's W. In cases where sphericity assumptions were violated, a Greenhouse–Geisser adjustment was applied. When the ANOVA tests demonstrated a statistically significant condition × time (× time spans or × percentage difference) interaction or a statistically significant main effect for condition, time, time spans, or percentage difference, post-hoc comparisons of the mean differences were performed using the Bonferroni correction ($p_{bonf}$). Non-significant interaction effects were excluded from the models prior to the examination of main effects. The variance explained by each ANOVA model is reported in $\eta^2$, and the mean difference effect sizes are reported as Cohen's d with 95% lower limit (LL) and upper limit (UL) confidence intervals [LL, UL].

## Sensitivity analysis

Due to the limited number of cadets at the Military department and the limited capacity of the sleep laboratory (6 participants per measurement at one time), the maximum number of participants was limited to 18. We performed sensitivity analyses to observed effects in our tests using G*Power (version 3.1.). For the within-factor differences and within-between interaction in repeated measures ANOVAs with α = 0.05, β = 0.8, N = 18, 2 conditions, and 5 testing

sessions, we reached a sensitivity to observe a Cohen's f = 0.267, which corresponds to $\eta^2$ = 0.066 (default values of correlation and sphericity of 0.5 and 1, respectively, were used).

For the post-hoc comparisons, we calculated the sensitivity for two-tailed t-tests of dependent means with matched pairs. With a Bonferroni-adjusted α of 0.01 (calculated as 0.05 divided by the number of comparisons, 5 in our case), β = 0.8 [48], N = 18, we reach sensitivity to observe an effect size (Cohen's d) of 0.894.

### Registration and supplementary materials

Project registration, associated dataset and JASP outputs for all performed analyses are available at the Open Science Framework [OSF] repository (URL: https://osf.io/bjecr; DOI: https://doi.org/10.17605/OSF.IO/BJECR).

The authors confirm that all ongoing and related trials for this drug/intervention are registered, including registration on ClinicalTrials.gov under the identifier number NCT05868798. The project was not preregistered as it was not a common practice in the field at the time of the project's development and data collection.

## Results

### Simple reaction time and incorrect trials

In simple reaction times, we observed statistically significant main effect of time ($F_{2.30, 6291.19}$ = 4.99, p = 0.009, $\eta^2$ = 0.19, with Greenhouse-Geisser correction). In subsequent post-hoc comparisons, the simple reaction time was statistically significantly slower at -12 compared to -24 (mean difference = 26.74 ms, $p_{bonf}$ = 0.002, Cohen's d = 1.05 [0.15, 1.94]) and + 8 (mean difference = -18.04 ms, $p_{bonf}$ = 0.007, Cohen's d = 0.70 [-1.51, -0.10]) (Fig 3A).

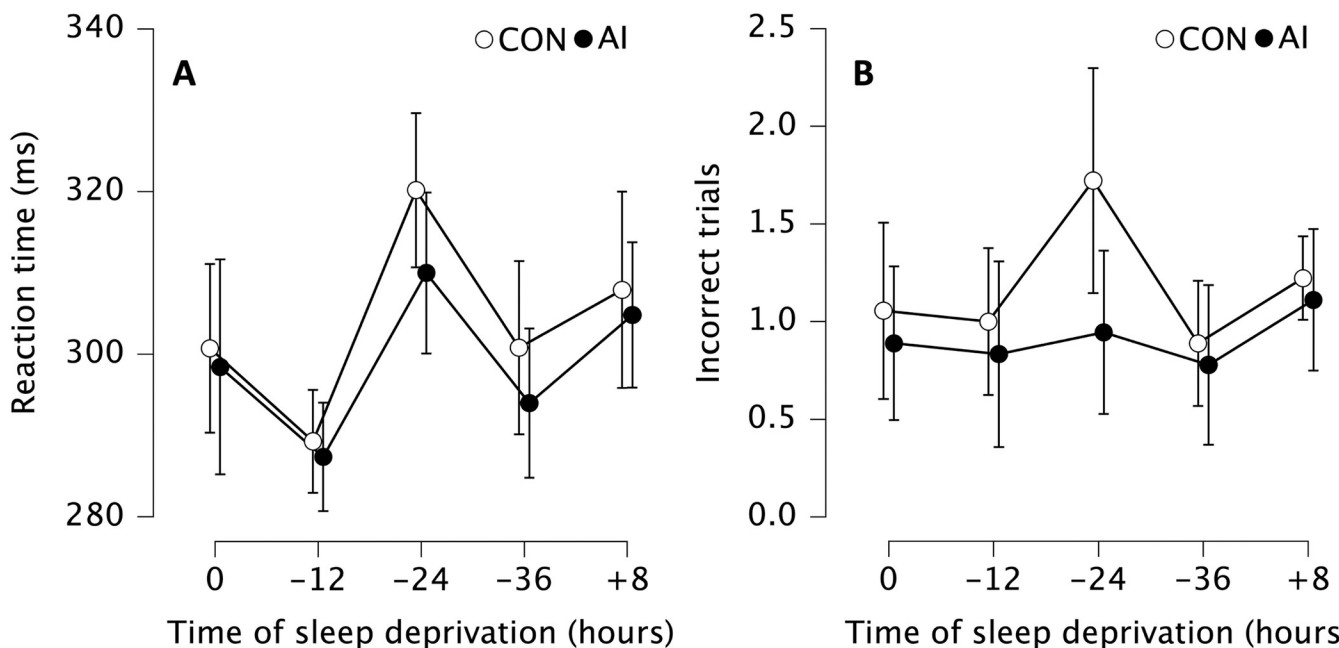

**Fig 3.** (A) Mean simple reaction time and (B) number of incorrect trials without (CON, white) and with ammonia inhalants (AI, black). Data are presented as mean (circles) and error bars represent their 95% confidence interval (LL, UL) at baseline (0 h), after 12 (-12 h), 24 (-24 h) and 36 (-36 h) hours of total sleep deprivation followed by 8 (+8 h) hours of recovery sleep.

There was also statistically significant main effect of condition ($F_{1,\ 1060.42}$ = 9.40, p = 0.007, $\eta^2$ = 0.014), demonstrating that application of AI reduced simple reaction time (mean difference = 4.85 ms, $p_{bonf}$ = 0.007, Cohen's d = 0.19 [0.04, 0.34]) (Fig 3A).

Though there was no statistically significant main effect of time ($F_{2.262,\ 2.619}$ = 1.437, $p_{bonf}$ = 0.250, $\eta^2$ = 0.051, with Greenhouse-Geisser correction) for incorrect trials execution, there was statistically significant main effect of condition ($F_{1,\ 3.200}$ = 6.800, $p_{bonf}$ = 0.018, $\eta^2$ = 0.028), demonstrating that using AI reduced the amount of incorrect trials (mean difference = 0.267, $p_{bonf}$ = 0.018, Cohen's d = 0.267 [0.04, 0.50]) (Fig 3B).

### Shooting accuracy

For shooting accuracy, we observed no statistically significant main effect of time ($F_{2.834,\ 35.844}$ = 1.755, $p_{bonf}$ = 0.171, $\eta^2$ = 0.045, with Greenhouse-Geisser correction) or main effect of condition ($F_{1,\ 27.222}$ = 1.529, $p_{bonf}$ = 0.233, $\eta^2$ = 0.012).

### Heart rate during shooting

We observed statistically significant main effect of time ($F_{4,\ 2959.875}$ = 8.515, $p_{bonf}$ < 0.001, $\eta^2$ = 0.126) on heart rate during shooting. In subsequent post-hoc tests, the heart rate was statistically significantly slower at 0 than at -12 (mean difference = 9.29 bpm, $p_{bonf}$ = 0.004, Cohen's d = 0.84 [0.15, 1.53]) and also slower at -24 compare to -12 (9.75 bpm, $p_{bonf}$ = 0.001, Cohen's d = 0.88 [0.17, 1.59]) and -36 (5.63 bpm, $p_{bonf}$ = 0.018, Cohen's d = 0.51 [0.09, 1.11]) (Fig 4A).

There was also a statistically significant condition × time span interaction ($F_{4,\ 212.765}$ = 22.593, $p_{bonf}$ < 0.001, $\eta^2$ = 0.009) (Fig 4B), demonstrating that after using AI we observed the highest heart rate at AI 0–15 compare to baseline (mean difference = 22.95 bpm, $p_{bonf}$ < 0.001,

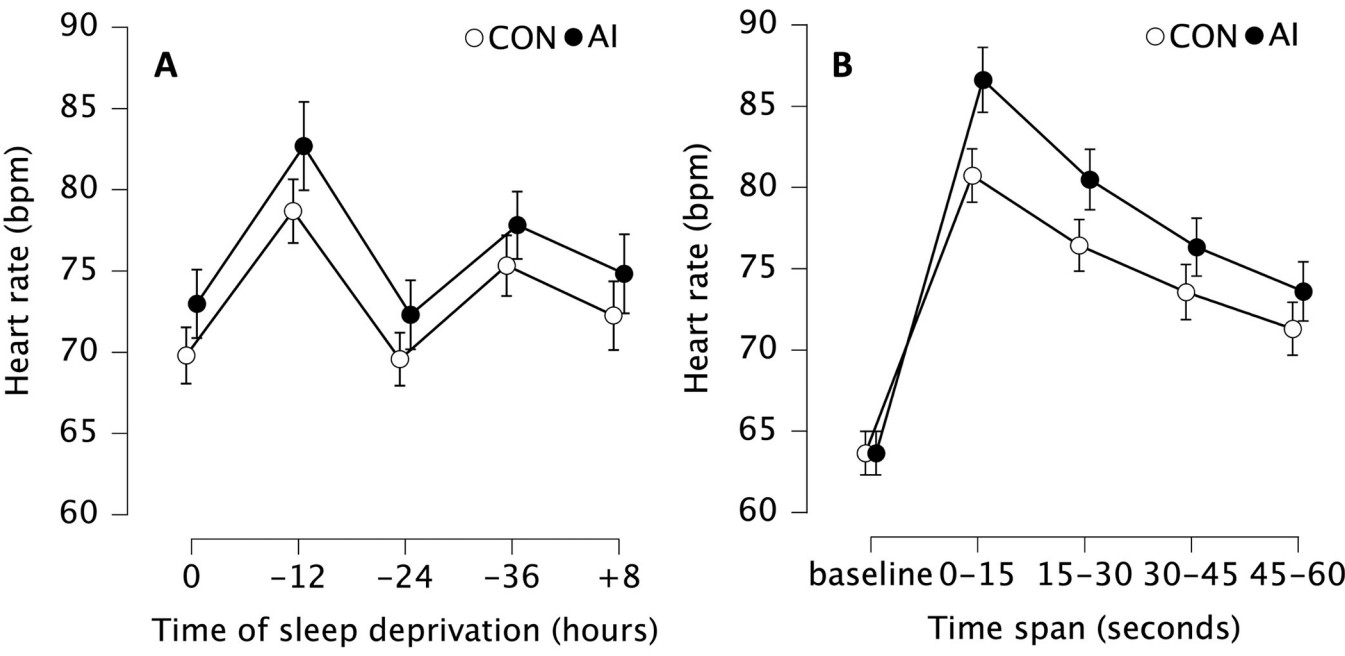

**Fig 4.** (A) Mean heart rate (bpm) without (CON, white) and with ammonia inhalants (AI, black). Data are presented as mean (circles) and error bars represent their 95% confidence interval (LL, UL) at baseline (0 h), after 12 (-12 h), 24 (-24 h) and 36 (-36 h) hours of total sleep deprivation followed by 8 (+8 h) hours of recovery sleep. (B) Mean heart rate (bpm) without (CON, white) and with ammonia inhalants (AI, black). Data are presented as mean (circles) and error bars represent their 95% confidence interval (LL, UL) as an average of 15-second time spans (0–15, 15–30, 30–45, and 45–60) during shooting trials regardless of sleep deprivation.

Cohen's d = 2.08 [0.68, 3.47]), AI 15–30 (mean difference = 6.13 bpm, $p_{bonf} < 0.001$, Cohen's d = 0.55 [0.09, 1.02]), AI 30–45 (mean difference = 10.28. bpm, $p_{bonf} < 0.001$, Cohen's d = 0.93 [0.26, 1.60]), and AI 45–60 (mean difference = 13.00 bpm, $p_{bonf} < 0.001$, Cohen's d = 1.18 [0.35, 1.99]). Without the use of AI we observed highest heart rate in CON 0–15 compare to baseline (mean difference = 17.07 bpm, $p_{bonf} < 0.001$, Cohen's d = 1.54 [0.49, 2.60]), CON 15–30 (mean difference = 4.29 bpm, $p_{bonf} < 0.001$, Cohen's d = 0.39 [0.01, 0.77]), CON 30–45 (mean difference = 7.16 bpm, $p_{bonf} < 0.001$, Cohen's d = 0.65 [0.14, 1.16]), and CON 45–60 (mean difference = 9.43 bpm, $p_{bonf} < 0.001$, Cohen's d = 0.85 [0.30, 1.45]).

Additionally, all heart rates in same time spans after AI administration were higher compared to CON. AI 0–15 was higher than CON 0–15 (mean difference = 5.88 bpm, $p_{bonf} < 0.001$, Cohen's d = 0.53 [0.13, 0.93]), AI 15–30 was higher than CON 15–30 (mean difference = 4.04 bpm, $p_{bonf} < 0.001$, Cohen's d = 0.36 [0.53, 0.68]), AI 30–45 was higher than CON 30–45 (mean difference = 2.76 bpm, $p_{bonf} < 0.001$, Cohen's d = 0.25 [0.01, 0.51]), AI 45–60 was higher than CON 45–60 (mean difference = 2.31 bpm, $p_{bonf} < 0.001$, Cohen's d = 0.21 [0.03, 0.45]) (Fig 4B).

Additionally, there was statistically significant main effect of percentage difference between AI and CON during the same time spans ($F_{3, 61.833} = 8.776$, $p_{bonf} < 0.001$, $\eta^2 = 0.340$), where we observed greater percentage difference at 0–15 than 30–45 (mean difference = 3.67%, $p_{bonf} = 0.012$, Cohen's d = 1.02 [0.18, 1.85]) and 45–60 (mean difference = 4.08%, $p_{bonf} = 0.005$, Cohen's d = 1.13 [0.27, 2.00]) (Fig 5).

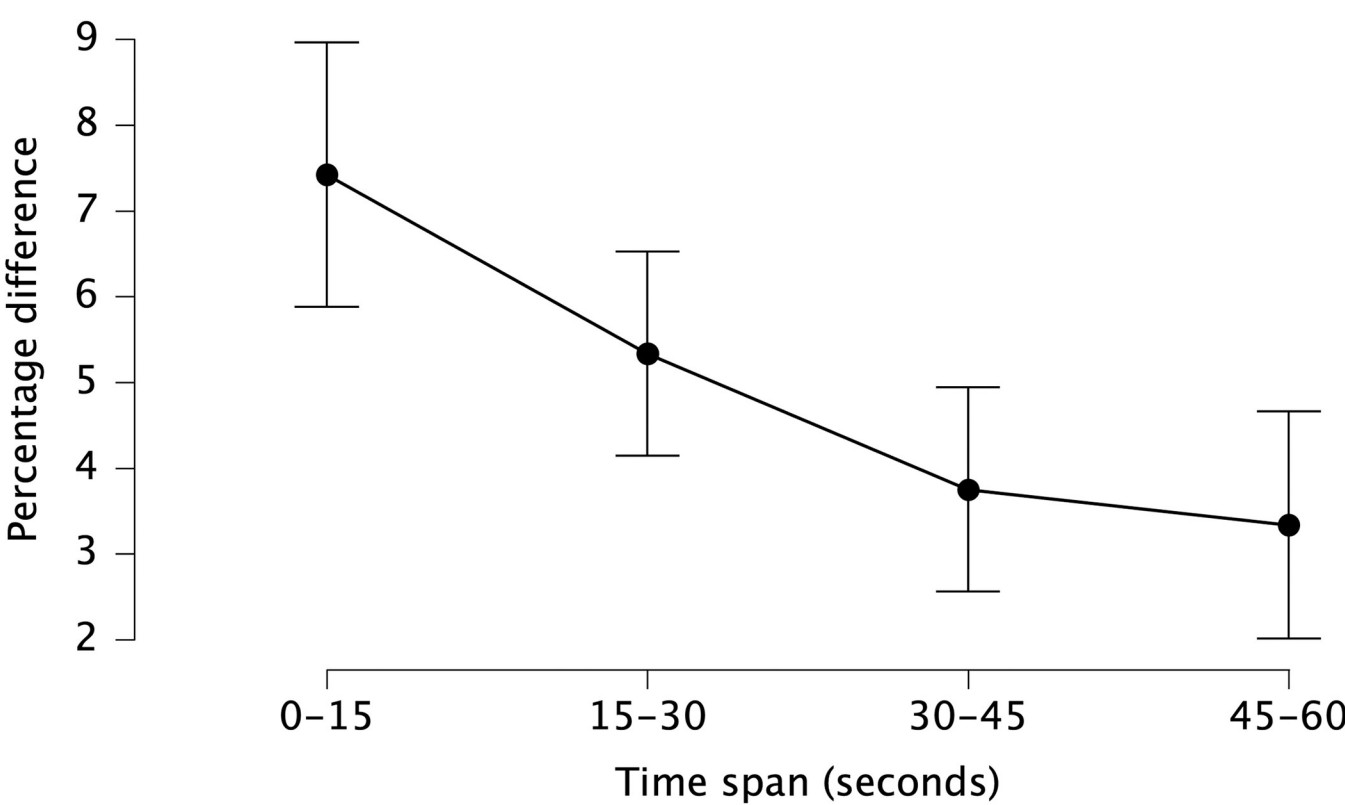

**Fig 5. Mean heart rate percentage difference (between CON and AI at each 15-second time span).** Data are presented as mean (circles) and error bars represent their 95% confidence interval (LL, UL) at the average of 15-second time spans (0–15, 15–30, 30–45, and 45–60) during shooting trials regardless of sleep deprivation.

## Countermovement jump height

In the case of jump height, we observed statistically significant main effect of time ($F_{4, 29.447}$ = 8.070, $p_{bonf}$ < 0.001, $\eta^2$ = 0.035). In subsequent post-hoc tests, the jump height was statistically significantly higher in the evenings compare to the mornings. Particularly -12 was higher than 0 (mean difference = 1.88 cm, $p_{bonf}$ = 0.015, Cohen's d = 0.33 [0.22, 0.65]), -24 (mean difference = 1.33 cm, $p_{bonf}$ = 0.010, Cohen's d = 0.24 [-0.05, 0.52]), +8 (mean difference = 1.35 cm, $p_{bonf}$ = 0.046, Cohen's d = 0.24 [-0.05, 0.53]), and also -36 was higher compare to 0 (mean difference = 2.03 cm, $p_{bonf}$ = 0.006, Cohen's d = 0.36 [-0.04, 0.67]) and -24 (mean difference = 1.50 cm, $p_{bonf}$ = 0.014, Cohen's d = 0.26 [-0.28, 0.56]) (Fig 6).

There was also a statistically significant main effect of condition ($F_{1, 18.142}$ = 10.576, $p_{bonf}$ = 0.005, $\eta^2$ = 0.035), demonstrating that jump height after AI administration was higher (mean difference = 0.64 cm, $p_{bonf}$ = 0.005, Cohen's d = 0.11 [0.03, 0.20]) compared to CON (Fig 6).

## Rating of perceived exertion and Epworth sleepiness scale

We observed statistically significant main effect of time ($F_{2.844, 10.030}$ = 10.478, $p_{bonf}$ < 0.001, $\eta^2$ = 0.282, with Greenhouse-Geisser correction) indicating that soldiers reported highest RPE at

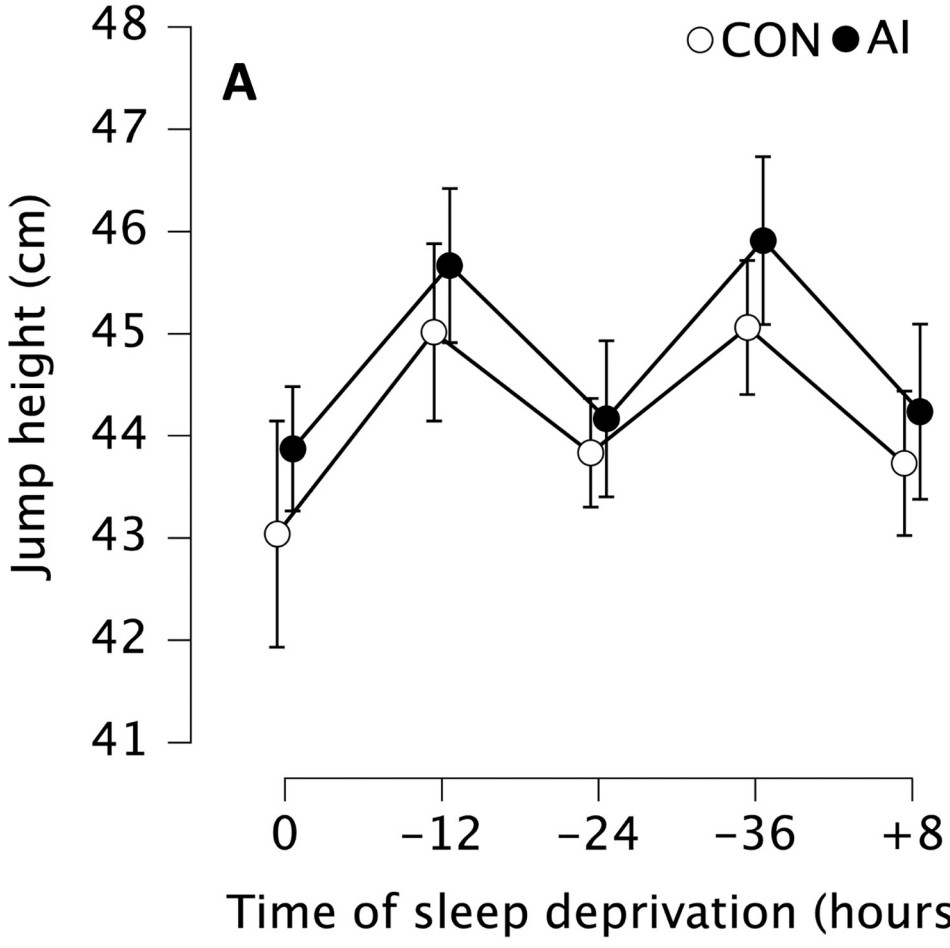

**Fig 6. Mean countermovement jump height without (CON, white) and with ammonia inhalants (AI, black).** Data are presented as mean (circles) and error bars represent their 95% confidence interval (LL, UL) at baseline after a full night of sleep (0 h), after 12 (-12 h), 24 (-24 h) and 36 (-36 h) hours of total sleep deprivation followed by 8 (+8 h) hours of recovery sleep.

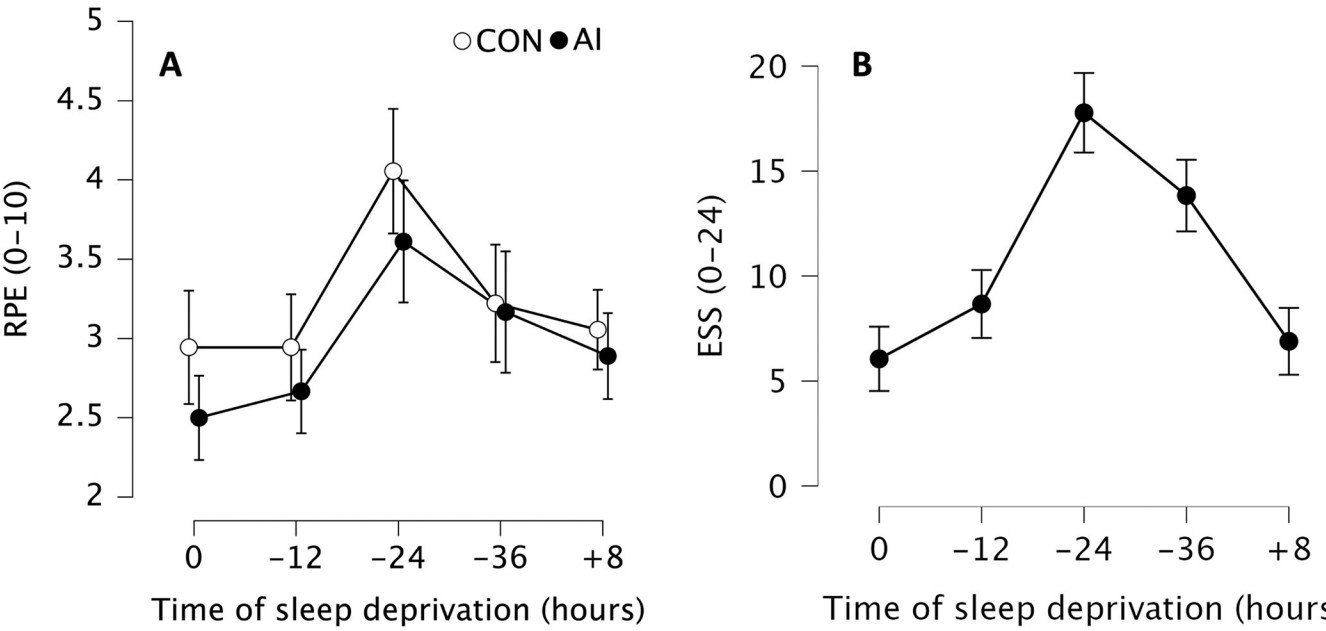

**Fig 7.** (A) Mean rating of perceived exertion (RPE) without (CON, white) and with ammonia inhalants (AI, black). Data are presented as mean (circles) and error bars represent their 95% confidence interval (LL, UL) at baseline after a full night of sleep (0 h), after 12 (-12 h), 24 (-24 h) and 36 (-36 h) hours of total sleep deprivation followed by 8 (+8 h) hours of recovery sleep. (B) Mean Epworth Sleepiness Scale (ESS) score. Data are presented as mean (circles) and error bars represent their 95% confidence interval (LL, UL) at baseline after a full night of sleep (0 h), after 12 (-12 h), 24 (-24 h) and 36 (-36 h) hours of total sleep deprivation followed by 8 (+8 h) hours of recovery sleep.

-24 compare to 0 (mean difference = 1.11, $p_{bonf} < 0.001$, Cohen's d = 1.19 [0.32, 2.06]), -12 (mean difference = 1.03, $p_{bonf}$ = 0.010, Cohen's d = 1.10 [-0.26, 1.93]), and +8 (mean difference = 0.86, $p_{bonf}$ = 0.002, Cohen's d = 0.92 [0.14, 1.70]).

There was also statistically significant main effect of condition ($F_{1,\ 3.472}$ = 17.220, $p_{bonf} < 0.001$, $\eta^2$ = 0.034), demonstrating that soldiers reported higher RPE at CON compared to after usage of AI (mean difference = 0.28, $p_{bonf} < 0.001$, Cohen's d = 0.30 [0.11, 0.48]) (Fig 7A).

For ESS, we observed statistically significant main effect of time ($F_{4\ 450.572}$ = 39.764, $p_{bonf} < 0.001$, $\eta^2$ = 0.701), indicating a statistically significantly greater sleepiness score at -24 compare to 0 (mean difference = 11.72, $p_{bonf} < 0.001$, Cohen's d = 2.91 [1.25, 4.58]), -12 (mean difference = 9.11, $p_{bonf} < 0.001$, Cohen's d = 2.26 [0.87, 3.65]), -36 (mean difference = 3.94, $p_{bonf}$ = 0.018, Cohen's d = 0.98 [0.03, 1.93]), and +8 (mean difference = 10.89, $p_{bonf} < 0.001$, Cohen's d = 2.70 [1.13, 4.28]) (Fig 7A).

Additionally, ESS was also statistically significantly greater at -36 compared to 0 (mean difference = 7.78, $p_{bonf} < 0.001$, Cohen's d = 1.93 [0.63, 3.20]), -12 (mean difference = 5.17, $p_{bonf}$ = 0.002, Cohen's d = 1.28 [0.25, 2.32]), and +8 (mean difference = 6.94, $p_{bonf} < 0.001$, Cohen's d = 1.72 [0.54, 2.91]) (Fig 7B).

## Rifle disassembly and reassembly protocol

For the sum of the rifle disassembly and reassembly time there was no statistically significant main effect of time ($F_{1,\ 1.362}$ = 0.098, $p_{bonf}$ = 0.759, $\eta^2$ = 0.0008) nor the main effect of condition ($F_{4,\ 10.825}$ = 0.754, $p_{bonf}$ = 0.560, $\eta^2$ = 0.027). Descriptions of all results can be seen in the following table (Table 1).

**Table 1. Descriptive statistics of all results at baseline after a full night of sleep (0 h), after 12 (-12 h), 24 (-24 h) and 36 (-36 h) hours of total sleep deprivation followed by 8 (+8 h) hours of recovery sleep (data are presented in mean ± SD, AI = ammonia inhalants, CON = without ammonia inhalants, RPE = rating of perceived exertion, ESS = Epworth Sleepiness Scale, DAS = rifle disassembly and reassembly protocol).**

| Task | Condition | N | Time of total sleep deprivation | | | | |
|---|---|---|---|---|---|---|---|
| | | | 0 | -12 | -24 | -36 | +8 |
| Simple reaction time (ms) | CON | 18 | 300.72 ± 21.13 | 289.3 ± 18.40 | 320.17 ± 18.40 | 300.81 ± 18.40 | 307.91 ± 18.40 |
| | AI | | 298.43 ± 26.40 | 287.38 ± 16.76 | 310.00 ± 30.95 | 293.99 ± 21.83 | 304.84 ± 24.36 |
| Shooting accuracy (points) | CON | 18 | 86.72 ± 3.33 | 86.72 ± 3.97 | 86.17 ± 4.02 | 85.89 ± 3.59 | 84.28 ± 4.66 |
| | AI | | 87.00 ± 5.41 | 87.28 ± 3.18 | 87.39 ± 3.65 | 86.33 ± 5.55 | 85.67 ± 4.86 |
| Heart rate (bpm) | CON | 18 | 72.06 ± 8.82 | 81.24 ± 13.43 | 71.95 ± 9.72 | 77.39 ± 12.80 | 74.84 ± 11.12 |
| | AI | | 76.05 ± 10.60 | 86.23 ± 15.84 | 75.38 ± 11.54 | 80.5 ± 12.50 | 78.06 ± 11.91 |
| Jump height (cm) | CON | 18 | 43.04 ± 5.75 | 45.01 ± 5.58 | 43.83 ± 5.64 | 45.06 ± 5.37 | 43.73 ± 4.96 |
| | AI | | 43.87 ± 5.52 | 45.67 ± 5.82 | 44.17 ± 5.02 | 45.91 ± 5.55 | 44.24 ± 5.35 |
| RPE (0–10) | CON | 18 | 2.94 ± 1.03 | 2.94 ± 0.97 | 4.06 ± 0.78 | 3.22 ± 1.13 | 3.06 ± 0.62 |
| | AI | | 2.50 ± 0.90 | 2.67 ± 0.94 | 3.61 ± 0.89 | 3.17 ± 1.12 | 2.89 ± 0.46 |
| ESS (0–24) | x | 18 | 6.06 ± 3.14 | 8.67 ± 4.63 | 17.78 ± 3.17 | 13.83 ± 4.54 | 6.88 ± 3.80 |
| DAS (sec) | CON | 14 | 33.18 ± 5.75 | 33.03 ± 4.95 | 33.69 ± 4.12 | 31.15 ± 3.01 | 31.71 ± 4.15 |
| | AI | | 32.84 ± 3.71 | 31.71 ± 4.25 | 32.21 ± 2.61 | 31.78 ± 4.35 | 33.23 ± 5.02 |

## Discussion

This study aimed to investigate the potential effect of AI usage on simple reaction time, shooting accuracy, countermovement jump height, and rifle disassembly and reassembly time during 36 h of total sleep deprivation, followed by 8 h of recovery sleep in military personnel. The principal findings of our investigation were that although both total sleep deprivation and AI affected some variables during the study protocol, AI inhalation did not have a great effect specifically during the period of sleep deprivation. Nevertheless, regardless of sleep deprivation, AI use increased heart rate from 0–15 seconds after inhalation, improved simple reaction time without increasing the number of errors, and increased countermovement jump height while simultaneously decreasing rating of perceived exertion. No changes in shooting accuracy or the rifle disassembly and reassembly time after AI were observed.

Acknowledging the potential risks of AI, it is crucial to mention that under certain conditions it can be toxic, potentially cause allergic reactions, mask concussive symptoms, and exacerbate breathing difficulties for those with respiratory issues. These risks emphasize that despite its common use for reviving fainting individuals in the U.S, the use of AI for enhancing athletic performance should be carried out under caution and professional supervision [29]. However, it is worth noting that we did not observe any adverse effects associated with its use during our research.

### Simple reaction time

**Effects of total sleep deprivation.** It has been documented that prolonged periods of total sleep deprivation can impair sustained vigilance, which may negatively impact cognitive performance [49, 50] and potentially impair military readiness. The present study found that simple reaction time (SRT) was slower (3.2–9.3%) in the mornings (following 24 h of sleep deprivation, and again after 8 h of recovery sleep) as compared to the evening (following 12 h of sleep deprivation). This is partly consistent with a previous reports [21, 51] showing that when straight wakefulness exceeds 16 h (typically the end of a normal day), SRT slowed down. These already slow SRTs became even slower as wakefulness was maintained throughout the night into the early morning hours leading up to 24 h of total sleep deprivation. However, the

current study observed that this phenomenon did not persist when total sleep deprivation reached a maximum of 36 h, and instead, SRT showed an improvement (5.9%) compared to the 24-h time point.

One potential explanation may be that cognitive performance is influenced by the interplay between the sleep homeostatic system (i.e., the biological drive for sleep) and the circadian rhythm (i.e., synchronization with the 24-h day/night cycle) [52]. These systems operate in a coordinated manner, following a 24-h sinusoidal pattern, with increased or decreased levels of fatigue depending on the time of day. As a result, total sleep deprivation can affect cognitive performance in a non-linear fashion, with the most pronounced cognitive impairments observed during the morning hours [53, 54]. Our results also align with previous research [55], where faster SRT was reported in the evenings (after 8–12 and 32–36 h of total sleep deprivation) compared to the morning (after 24 h of total sleep deprivation). This phenomenon is also known as the "wake maintenance zone", where the desire to sleep is usually the lowest in the evenings (2 to 3 h before the start of the melatonin secretion), and it is also maintained despite the increasing drive for sleep that results from total sleep deprivation [56].

**Effects of ammonia inhalants.**   Even though SRT was slower in the mornings as compared to the evenings, our study aimed to determine whether using AI would affect SRT during total sleep deprivation. Partly in line with our predictions, SRT was faster following the use of AI (1.6%), but AI had no greater effect on SRT during sleep deprivation than when fully rested. Regardless of sleep deprivation, these ergogenic effects of AI are thought to be mediated by the activation of the olfactory and trigeminal nerves, which leads to the activation of the adrenergic receptor and the subsequent release of norepinephrine, resulting in an increase in cardiac output and respiratory rate [29]. This effect has been shown to result in an increase in beat-to-beat middle cerebral artery blood flow velocity [31], which may be linked to enhanced cognitive performance [57].

Hence, it is plausible that the increase in alertness following the use of AI may be attributed to improved delivery of oxygenated blood to the brain [58]. Previous studies have indicated that decreased cerebral blood flow can impair cognitive function [59], which may result in reduced arousal levels, ultimately leading to decreased performance [60]. Considering this evidence, the enhancement of alertness [32, 35] resulting from AI use may explain the widespread use of this stimulant. Nevertheless, it should be noted that the implications of these findings are beyond the scope of this study.

Despite this, our results suggest that the utilization of AI may be beneficial for soldiers, as it may improve their SRT without increasing the number of errors, regardless of the level of total sleep deprivation. In military contexts, where performance demands are high and have significant implications for many individuals' well-being, SRT can be a critical determinant of success or failure and may even have life-or-death consequences [61]. To sum up, even though it must be taken into account that the difference we observed was on the order of milliseconds, the utilization of AI may have potential applications in specific military scenarios, no matter the length of sleep deprivation.

## Cardiovascular response

**Effects of total sleep deprivation.**   It is widely acknowledged that the autonomic nervous system influences heart rate (HR) through the discharges of the sympathetic and the parasympathetic nervous system via sympathetic and vagal innervations, respectively [62]. However, evidence regarding the effect of total sleep deprivation on these systems, remains inconsistent [62]. One study reported [63] that after 30 h of total sleep deprivation, there was a decrease in cardiac sympathetic activity, but no change in parasympathetic, which contradicts the findings

of another study [64] that found increased sympathetic activity and decreased parasympathetic activity following 36 h of total sleep deprivation. Despite the inconsistent findings of previous studies, our results indicated that HR during shooting protocol was lower after 24 h of total sleep deprivation, with a slight increase after 36 h, without considering the effects of AI. These results partly align with the outcomes of laboratory studies that controlled for environmental and behavioral influences such as sleep, light and activity, reporting that 24–30 h of total sleep deprivation results in a decrease in HR in healthy young individuals [65–67]. This decline in HR is typically superimposed on a 24-h rhythm [63], which could be a potential explanation for the increase in HR after 36 h of sleep deprivation in the current study.

**Effects of ammonia inhalants.**   Nevertheless, this study also investigated the impact of AI inhalation on HR during shooting protocol while total sleep deprived. Our findings are in line with a previous study [31], which demonstrated that AI elicits a strong cardiovascular response that results in an elevation of HR compared to control trials without AI. Additionally, our results showed that this response was observed immediately after AI inhalation, with HR increasing in the first 15 seconds (7.3% compare to CON) and then gradually declining (Fig 5). That is consistent with prior reports indicating that the effect of AI on the cardiovascular system is typically short-term and may not provide a sustained ergogenic benefit beyond the initial 15 seconds of inhalation [29, 31].

In sum, despite the consistency of our findings with those of the prior investigation, it is imperative to consider that the regulation of HR is predominantly controlled by the autonomic nervous system and the opposing actions of its sympathetic and parasympathetic components. This constitutes a complex system that may be impacted by various confounding factors during the prolonged experiments, as were conducted in this study, such as dietary intake, energy balance, hydration, physical activity, effect of shooting stress, temperature regulation, and psychological stress. Thus, the following potential studies should consider testing cardiovascular responses throughout the whole study period.

**Shooting accuracy.**   Assuming that inhaling AIs could be associated with higher arousal [29], we had predicted that AI could attenuate the shooting accuracy decrements that would be caused by sleep deprivation [68]. However, in the context of the present study, neither sleep deprivation nor the presence of AI (i.e., increased HR) affected shooting accuracy. The results of our study partially concur with previous research that found that the live-fire shooting accuracy of trained soldiers was not impacted by total sleep deprivation ranging from 24 to 36 h [69, 70]. However, it has been reported that similarly extended total sleep deprivation resulted in impaired live-fire shooting accuracy among conventional military personnel and non-military trained individuals [15, 71, 72]. This possibly suggests that more experienced shooters (with longer training experience) may be more resilient to the possible adverse effects of total sleep deprivation, which may explain the consistent shoot accuracy found in the present study.

Despite these fairly straightforward findings, some observations in our study diverged from previous findings. For example, one study shows that shooting accuracy decreased as HR increased in standing positions [73], but another study suggests that low-intensity exercise, which leads to commensurate increases in HR, may initially enhance shooting accuracy before a decline is observed. Nonetheless, the different outcomes observed could be attributed to the substantial intra-individual variability in managing psychological factors, such as fatigue and stress, which may impact HR and, consequently, shooting accuracy among individuals [74]. Therefore, while our results agree with previous research [16, 72] that also evaluated shooting accuracy using small arms simulators rather than live fire, it is crucial to acknowledge that the use of simulated weapons in evaluating shooting performance may affect the applicability of the findings to real-life military situations [74].

In the present study, it is challenging to differentiate the various effects of HR as an indicator of physiological exertion or mechanical perturbation. This intra-individual variability may be attributed to the heightened mechanical impact of the heartbeat on shooting dynamics or may indicate an "over-arousal" that results in a decrease in performance, as demonstrated by the Yerkes-Dodson performance-arousal curve [75]. With all these points in mind, further research should delve into the interaction between these mechanical, physiological, and psychological factors that influence shooting more deeply to better understand the interplay of these elements. In summary, this study highlights that although AI did not improve handgun shooting accuracy, it was posed no negative effect either. It is still important to keep in mind the parameters of our study (observed effect sizes). However, the unaffected shooting accuracy, combined with the decreased SRT noted in the previous section, can be relevant in real-life military contexts where a soldier may need to react quickly and shoot accurately.

**Rifle disassembly and reassembly.** Many occupations that require prolonged periods of wakefulness, including soldiers, also demand the ability to maintain manual dexterity. Previous research has indicated that impaired manual dexterity can be observed after 24 h of total sleep deprivation among medical professions [76]. As previously discussed, while AI has been assumed to be an effective countermeasure for preserving vigilance and attention [29], concerns have arisen about its potential adverse effects on tasks involving manual dexterity, particularly in terms of increased tearfulness or hand tremors—a typical reaction to inhalations of AI. However, in our results we did not observe any impact on rifle disassembly and reassembly time as a result of total sleep deprivation or the usage of AI. The current findings align with the results of previous studies that have been conducted on military personnel, demonstrating that a 24 h period of total sleep deprivation did not impact the manual dexterity in rifle disassembly and reassembly [77] or the laboratory-based Grooved Pegboard test [78].

The disparities between studies outcomes may likely be attributed to the various methods used for measuring manual dexterity in medical personnel, such as the virtual laparoscopy simulator, or to the possibility of movement automation that soldiers may develop as a result of frequent repetition of tasks during their service, referred to as a "drill." Although the present study did not specifically evaluate variables as hand steadiness, but only total time to finish a task, collectively, the evidence suggests that neither total sleep deprivation nor usage of AI has an impact on manual dexterity in military personnel.

## Countermovement jump height and rating of perceived exertion

**Effects of total sleep deprivation.** The presented study found that the mean countermovement jump (CMJ) height was higher among participants during the evenings as compared to all morning measurements (3.0–4.7%). The observed outcome aligns with the circadian rhythm reported in the preceding sections. Specifically, the pattern of results associated with total sleep deprivation adheres to expectations, whereas the morning session conducted after 24 h of total sleep deprivation (a point where participants were reported to be most tired based on ESS scores) does not yield worse results than those obtained after 0 h, and following 8 h of recovery sleep. Additionally, evening sessions conducted after 36 h of total sleep deprivation did not yield worse outcomes than those obtained after 12 h (Fig 5A) This finding can be attributed to the increased performance in the CMJ, which demonstrates a clear circadian rhythm, with superior outcomes recorded during the afternoon relative to the morning, as described in previous research studies [79, 80]. Total sleep deprivation has been suggested to affect short-term anaerobic performance, such as jumping, by reducing motivation [81] and increased mental fatigue [82]. However, this study did not collect any of such measures, leaving this interpretation of results as speculative.

Based on previous studies [20, 83], total sleep deprivation may also lead to changes in the perception of effort. The RPE increased after 24 to 30 h of total sleep deprivation alongside a decrease in performance [20, 83]. These previous observations partly align with the results of the present study, where the highest RPE was reported after 24 h of total sleep deprivation (Fig 7A), which also corresponded with the highest level of subjective sleepiness reported by our participants (Fig 7B). Although our participants reported feeling less tired at 36 h than after 24 h (28.6%) of total sleep deprivation, this apparent "improvement" should not be interpreted as an actual improvement, but it is rather a continuation of the circadian rhythm. Eventually, the night of recovery sleep seemed to more or less returned the perception of sleepiness to levels in line with the 0 h of total sleep deprivation and normal circadian rhythm.

**Effects of ammonia inhalants.** Our results also suggest that using AI may enhance performance in CMJ height by 1.5% and concurrently reduces RPE by 9.4%, regardless of total sleep deprivation. Notably, the observed improvements may be attributed to the psychological arousal effects of AI, such as heightened alertness, which have been documented in previous research [32, 35]. This phenomenon commonly referred to as the "psyching-up" effect of AI, may enhance short-term performance by activating the sympathetic branch of the autonomic nervous system and increasing psychological activation [29]. Based on that, previous studies showed an increased peak rate of force development during the isometric mid-thigh pull exercise [31, 33], an explosive isometric strength task using a slow stretch-shortening cycle, the same as during CMJ. Additionally, the usage of AI increased power output over repeated high-intensity sprint exercises [35]. Regarding the following facts, while assuming a correlation between CMJ performance and short sprints [84], the usage of AI may have potential applications in some specific military scenarios, but more studies delineating optimum protocols and delivery methods are necessary.

Although previous studies have investigated the effects of total sleep deprivation and AI independently, none included the combination of these conditions. Furthermore, these studies all used various measuring techniques (CMJ with arm movement vs without; peak jump height vs average jump height; different rest periods between jumps; estimated power output vs direct assessment, etc.), all of which can impact the data, decreasing comparativeness of their findings to ours. Therefore, we acknowledge that the data from those studies bring interesting insights, but more studies with consistent methodology are still needed in order to make justifiable comparisons.

To sum up, when considering our data and the data from previous studies collectively, AI may positively impact explosive performance, regardless of the individual's sleep status. Our results also indicate that this effect may have some transfers to real-world dynamic exercises, such as jumping performance. Consequently, even though a decrease in RPE after the physical performance and improvement in short-term movements, such as jumping over a barrier or sprinting across a battlefield, may even have life-or-death consequences in military contexts, it is still necessary to take these results with caution. There is currently a shortage of published research examining the effect of AI on performance, and further research in this area is necessary to establish a more robust evidence base regarding its effect.

## Limitations

This study's strengths comprise the tightly controlled protocol and crossover randomized controlled trial design, which allowed for direct within-subject repeated-measures investigation of the effects of total sleep deprivation and ammonia inhalation. However, study is not without limitations. For example, we were not able to perform any military-specific physically demanding tasks (e.g., casualty drag, wall climb, sprinting with personal protective equipment, loaded

carries, etc.) because of limited space in the laboratory-based setup. Nevertheless, the CMJ is a commonly used physical task to assess explosive neuromuscular performance, and the linear position transducer that we used is a reliable and space friendly tool for assessing CMJ. Therefore, as AI increased CMJ height, increased HR, and improved SRT in our study, future researchers should investigate the effects of AI on more demanding military-specific tasks (or tasks that last longer than the CMJ) where increasing HR or improving SRT may help aid performance. Additionally, one limitation of this study was the absence of a placebo condition due to the distinct characteristics of AIs. Participants can likely distinguish AIs from any potential placebo due to their strong smell and immediate physical response after inhalation. Though some placebo substances like menthol oil or Vicks VapoRub have been used in other studies [29], these also have identifiable (though arguably less intrusive) smells that can potentially compromise the blinding process. Consequently, participants in our study likely knew when they were and were not inhaling an AI. Nonetheless, no studies have reported an increase in performance superior to AI inhalation for placebo or control conditions, indicating a limited placebo effect in this context.

Additionally, we sourced our participants from a cohort of cadets at the Military department, which also has its own capacity constraints. To maintain the standardization of the study, we aimed to test participants with similar daily routines. We specifically sought to include participants within a short period around the equinox to account for any seasonal variations that could impact the circadian rhythms. This intention was to minimize the potential influence of individual lifestyle differences and external factors related to seasonal changes on the study results.

Furthermore, our study encountered some logistical constraints that determined the number of participants we could feasibly accommodate. Our sample size was limited to 18 participants due to the maximal capacity of the sleep laboratory we utilized for this research. The laboratory could only accommodate six subjects in one testing session, which created a practical ceiling on the number of individuals who could participate simultaneously.

Despite the relatively small sample size, we believe the homogeneity of the participants and the tightly controlled conditions under which the study was conducted add some robustness to the reliability of our findings. Nevertheless, we recognize the benefit of a larger and more diverse sample size in future studies to enhance the generalizability of our findings.

## Conclusion

Overall, despite the lack of reduction in the adverse effects of total sleep deprivation, the use of AI was found to cause a short-term increase in HR and enhance SRT without increasing the number of errors and increase CMJ height while concurrently decreasing the RPE. However, no changes were observed in handgun shooting accuracy and manual dexterity performance. These results indicate potential applications of AI in specific military scenarios, regardless of the presence or the length of total sleep deprivation. Nevertheless, caution is warranted in generalizing these results, given the study's limitations, such as the small, all-male sample size. Future research in this field would benefit from including more diverse participant groups and employing a wider range of military-related performance tests. This would provide a more comprehensive understanding of AI's effects under conditions of sleep deprivation, informing its potential utility in various operational contexts.

## Supporting information

**S1 Checklist. Consort checklist.**
(DOCX)

**S2 Checklist. Clinical studies checklist.**
(DOCX)

**S1 File. Institutional Review Board (IRB) approval original.**
(PDF)

**S2 File. Institutional Review Board (IRB) approval translated.**
(PDF)

**S3 File. Trial study protocol original.**
(DOCX)

**S4 File. Trial study protocol translated.**
(DOCX)

## Acknowledgments

We would like to extend our sincere gratitude to the participants who took part in this study.

**Disclaimer:** The views expressed are solely those of the authors and do not reflect the official policy or position of the Czech Army, the Department of Defense, or the Czech Government.

## Author Contributions

**Conceptualization:** Jan Maleček, Dan Omcirk, Tomáš Větrovský, Zdeňka Bendová, James J. Tufano.

**Data curation:** Jan Maleček, Dan Omcirk, Kateřina Skálová, Jan Pádecký, Martin Tino Janikov, Michael Obrtel, Michal Jonáš, David Kolář, Vladimír Michalička, Karel Sýkora.

**Formal analysis:** Jan Maleček, Tomáš Větrovský, Vít Třebický.

**Funding acquisition:** Jan Maleček, Michal Vágner, James J. Tufano.

**Investigation:** Jan Maleček, Dan Omcirk, Kateřina Skálová, Jan Pádecký, Martin Tino Janikov, Michael Obrtel, Michal Jonáš, David Kolář, Vladimír Michalička, Karel Sýkora, Michal Vágner, Lubomír Přívětivý, Tomáš Větrovský, Zdeňka Bendová, James J. Tufano.

**Methodology:** Jan Maleček, Dan Omcirk, Tomáš Větrovský, James J. Tufano.

**Project administration:** Jan Maleček, Dan Omcirk, Kateřina Skálová.

**Resources:** Jan Maleček, Kateřina Skálová, Lubomír Přívětivý.

**Software:** Jan Maleček.

**Supervision:** Jan Maleček, Dan Omcirk, Kateřina Skálová, Zdeňka Bendová, James J. Tufano.

**Validation:** Jan Maleček, Tomáš Větrovský, Vít Třebický, James J. Tufano.

**Visualization:** Jan Maleček, Tomáš Větrovský, Vít Třebický, James J. Tufano.

**Writing – original draft:** Jan Maleček.

**Writing – review & editing:** Jan Maleček, Dan Omcirk, Kateřina Skálová, Jan Pádecký, Martin Tino Janikov, Michael Obrtel, Michal Jonáš, David Kolář, Vladimír Michalička, Karel Sýkora, Michal Vágner, Lubomír Přívětivý, Tomáš Větrovský, Zdeňka Bendová, Vít Třebický, James J. Tufano.

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
