## [Decision Letter · Decision Letter 0]

6 Jul 2023

PONE-D-23-07437Effects of 36 hours of sleep deprivation on military-related tasks: can ammonium inhalants maintain performance?PLOS ONE

Dear Dr. Malecek,

Thank you for submitting your manuscript to PLOS ONE. After careful consideration, we feel that it has merit but does not fully meet PLOS ONE’s publication criteria as it currently stands. Therefore, we invite you to submit a revised version of the manuscript that addresses the points raised during the review process.

The manuscript has been evaluated by one reviewer and a member of our statistical advisory board, and their comments are available below. They have raised a number of concerns that need attention. They request additional information on methodological aspects of the study, revisions to the statistical analyses and revisions to the Discussion to ensure that the conclusions are presented appropriately.

Could you please revise the manuscript to carefully address the concerns raised?

Please note that we have only been able to secure a single subject reviewer to assess your manuscript. We are issuing a decision on your manuscript at this point to prevent further delays in the evaluation of your manuscript. Please be aware that the editor who handles your revised manuscript might find it necessary to invite additional reviewers to assess this work once the revised manuscript is submitted. However, we will aim to proceed on the basis of this single review if possible. 

We look forward to receiving your revised manuscript.

Kind regards,

Marianne Clemence

Staff Editor

PLOS ONE

Journal Requirements:

2 We note that the grant information you provided in the ‘Funding Information’ and ‘Financial Disclosure’ sections do not match.

3. Please include your tables as part of your main manuscript and remove the individual files. Please note that supplementary tables (should remain/ be uploaded) as separate "supporting information" files.

4. We note that the original protocol that you have uploaded as a Supporting Information file contains an institutional logo. As this logo is likely copyrighted, we ask that you please remove it from this file and upload an updated version upon resubmission.

Reviewers' comments:

Reviewer's Responses to Questions

**Comments to the Author**

1. Is the manuscript technically sound, and do the data support the conclusions?

Reviewer #1: Yes

Reviewer #2: Yes

2. Has the statistical analysis been performed appropriately and rigorously? 

Reviewer #1: Yes

Reviewer #2: No

3. Have the authors made all data underlying the findings in their manuscript fully available?

Reviewer #1: Yes

Reviewer #2: Yes

4. Is the manuscript presented in an intelligible fashion and written in standard English?

Reviewer #1: Yes

Reviewer #2: Yes

5. Review Comments to the Author

Reviewer #1: General comments

This paper investigated the effect of ammonia inhalants (i.e. a fast-acting ergogenic aid) on cognitive and physical performance (i.e. 5 tests repeated in 5-time points) during a night of total sleep deprivation (i.e. 36 hours of sleep deprivation) in young military personnel (i.e. 18 male cadets).

Authors found that there was no condition and time interaction in any test, but there was faster Reaction Time without increasing the number of errors, higher Jump height, lower RPE, and higher Heart Rate after using Ammonia Inhalants compared to Control regardless of Total Sleep Deprivation.

Overall, the paper is appropriately organized, well-written, and easy to follow. However, I do have many concerns, especially about the methods employed (familiarization, sleep control …), the study limitations (the sample size, blinding …), and the conclusions (repeating results …). I have listed these concerns in detail below.

Specific comments

Abstract

- Line 79 (also Line 173): I suggest replacing “cadets” because it is with double meaning, or explaining it in parentheses (“young armed”, “police force”) to avoid confusion.

- L86: “CON” should be preceded by “control” because it is a new abbreviation

- L88, 89: I suggest using p < 0.01

- L99: I suppose that “short-term increase in HR” should not be stated in the conclusion because it is not a performance improvement (we focus only on important results about the subject: impact of AI on performance during TSD at least in the conclusion)

- L101: I suggest that “the no changes” should not be stated in the conclusion

- L101, 102: The conclusion should be tempered and corrected; “AI could considerably improve performance in similar conditions and tests regardless of TSD (and not any duration of SD)”

- L103: keywords are not well selected; I suggest adding Performance (not tests), Ammonia Inhalants (not Ergogenic Aids), Soldiers or Army (not both), and Sleep Deprivation (not sleep loss).

Introduction

- L104: I suppose that there is a lack of general introduction concerning the disrupted sleep-wake cycle in everyday life (stress, shift work, jetlag…) before moving on to the part that specifies its relation with the military.

- L105: I suggest adding another more general scientific definition of “sleep” before this specific definition.

- L116, 117: The sentence should be reformulated; I suggest this correction; to various “physical”, cognitive, and behavioral impairments, such as fatigue and “decreased” attention and reaction time.

- L136: I suggest using “psycho-stimulant” rather than “stimulant” (more accurate)

-L139: I suppose that this study is very interesting about this topic and it could be added as a recent reference (Repeated low-dose caffeine ingestion during a night of total sleep deprivation improves endurance performance and cognitive function in young recreational runners: A randomized, double-blind, placebo-controlled study (2022))

https://doi.org/10.1080/07420528.2022.2097089

-L143: “The potential positive effects of caffeine are likely to manifest about 1 hour after ingestion relative to many reviews and meta-analysis (and not between 10 and 60 min) (Reference: Grgic J, Grgic I, Pickering C, et al. Br J Sports Med 2020;54:681–688)

-L156: I suppose that the duration effect is also an important thing. Is there any information about how long it last for AI (the half-life of caffeine lasts between 2.5 and 4.5 h in humans)?

-L163: I wonder if there are negative effects of AI in the literature (you should report in addition to the test results if there are any observed negative effects (during the experiments) of this stimulant in the discussion section).

-L164-168: This paragraph should clearly describe the objectives of the study: TSD (36 h), cognitive (SRT…) physical (JH) military personnel (18 male).

Methods

-L171: The VO2max is also an important measurement (if it was done, add it)

-L173: As outlined before, “cadets” need to be explained in parentheses (“young armed”, “police force”) to be clearer for readers.

-L184: The participants' sleep quality (and quantity) is a very important measurement for such a study. For example, the sleep quality of athletes could be determined using the questionnaire of Horne and Östberg (1976). This latter could inform about the participants rising times, bedtimes, and sleep disorders.

-L192: I suppose that the time should be written in a scientific article: 18:00 h and not in the military version: 1800 h (to be replaced throughout the manuscript)

-L196: Is there a familiarization (before the testing day) with the sleep deprivation (36 hours) to be familiar with the experimental process and the sleep adaptation? If yes, describe this familiarization.

-L202: Following “from”, we should put “to”.

-L202: How to be sure that participants have really passed 8 hours of recovery sleep? In the same context; how to be sure that the circadian rest-activity rhythms of the 18 subjects included in the study are well synchronized? It is obvious that only the use of devices for recording activity (actimeter, polysomnograph) or temperature, which are considered marker rhythms, could confirm the synchronization. Unfortunately, this was not carried out in this study.

-L210: I suggest correcting as follows: “either with ammonia inhalants (AI) or with control (CON)”

-L213-217: The experimental protocol was not blinded in this study. This latter could influence the performance outcomes (participants could be more motivated with AI supplement) which could limit the study conclusions. This should be added in the section of the study limitations.

-L224: I wonder if the mean energy intake per capita/day was calculated for participants. If yes, please add it.

-L252: It is useless to rewrite “in a randomized order” because it was already stated in the methods section. (To be corrected throughout this section)

-L258: I suggest starting by giving the quality measured by the test (accuracy, speed, strength) and the type of test (cognitive, physical, psychological), before the description, and this for all 5 tests (for more clarity).

-L319-323: The familiarization information should be written in the methods section. In this section, we only describe the realization of the test.

-L342: All abbreviations should be described; (LL, UL)

Results

-L364: I suggest replacing “incorrect trials” with “incorrect detections” for more clarity.

-L379: I also suggest replacing “condition” with “ammonia”, which would be clearer and easier to follow for readers.

-L390-443: There is no need to present all the results especially when they are unnecessary (or not important for the conclusions) and will bother readers.

Discussion

- I suggest that this section should be rewritten in a more precise way. It should focus on essential results about the main subject: the impact of TSD on the tests selected, and the impact of AI on performance tests (by describing the scientific reasons about previous similar works, without repeating results or useless information).

-L516: This reference is important and could also be added; Amir Khcharem, Makram Souissi, Rim Atheymen, Lobna Ben Mahmoud & Zouheir Sahnoun (2020): Effects of caffeine ingestion on 8-km run performance and cognitive function after 26 hours of sleep deprivation, Biological Rhythm Research, DOI: 10.1080/09291016.2020.1778975

Limitations

-L705: Some limitations of the study were not stated although they could influence the conclusions; the sample size is quite small; 18 and only men were used), the experimental protocol was not blinded, and qualitative and quantitative evaluation of sleep were not carried out.

Conclusion

I suggest that the conclusion needs to be reformulated; there is no need to repeat the results in this section and with the same abbreviations, (e.g. ammonia inhalant markedly improved the reaction time, jumping, and sensation of pain of soldiers). It should conclude the main results in a good manner. In addition, it should be tempered (e.g. in similar testing conditions of prolonged sleep deprivation). It should also finish with a recommendation for future studies about this topic (e.g., Further investigations including females and using different tests are needed…)

-L724: I suppose that the sentence “No matter the length of sleep deprivation” was inaccurate and should be replaced.

Reviewer #2: A single cohort study of 18 healthy male cadets investigated the acute effects of ammonia inhalants (AI) versus control on cognitive and physical performance during a 36 hour period of total sleep deprivation (TSD). ANOVA methods were used to analyze the main objectives of the trial. The conclusions are unclear.

Major revision:

If the interaction effect is significant, provide an interpretation of the results, but do not test main effects because the tests for main effects are uninteresting in light of significant interactions. If interaction effects are non-significant, drop the interaction effects from the model and test the main effects. Determining which results to present when testing interactions is often a multi-step process. Typically these type of results are presented in table format because they are more easily comprehendable.

6. PLOS authors have the option to publish the peer review history of their article (what does this mean?). If published, this will include your full peer review and any attached files.

Reviewer #1: **Yes: **Khcharem Amir

Reviewer #2: No

---

## [Author Response · Author response to Decision Letter 0]

24 Aug 2023

Dear Editor,

We are deeply appreciative of the insightful comments provided on our manuscript. The reviewers' constructive feedback was instrumental, guiding us in refining and enhancing our work. After incorporating these suggestions, we believe the manuscript has become significantly stronger, more comprehensive, and better positioned to contribute meaningfully to the field.

Reviewer #1: General comments

This paper investigated the effect of ammonia inhalants (i.e. a fast-acting ergogenic aid) on cognitive and physical performance (i.e. 5 tests repeated in 5-time points) during a night of total sleep deprivation (i.e. 36 hours of sleep deprivation) in young military personnel (i.e. 18 male cadets).

Authors found that there was no condition and time interaction in any test, but there was faster Reaction Time without increasing the number of errors, higher Jump height, lower RPE, and higher Heart Rate after using Ammonia Inhalants compared to Control regardless of Total Sleep Deprivation.

Overall, the paper is appropriately organized, well-written, and easy to follow. However, I do have many concerns, especially about the methods employed (familiarization, sleep control …), the study limitations (the sample size, blinding …), and the conclusions (repeating results …). I have listed these concerns in detail below.

Thank you for your valuable feedback on our manuscript. We appreciate your detailed comments regarding the methodology, study limitations, and conclusions. We acknowledge the importance of these issues and commit to making the necessary modifications in our revised manuscript. We believe the inclusion of your suggestions enhanced the rigor of our work.

Specific comments

Abstract

- Line 79 (also Line 173): I suggest replacing “cadets” because it is with double meaning, or explaining it in parentheses (“young armed”, “police force”) to avoid confusion.

Thank you for your insightful comment regarding the use of the term "cadets". We have added the word "military" before "cadets" throughout the manuscript. 

- L86: “CON” should be preceded by “control” because it is a new abbreviation

Thank you for your keen observation, we have added "as control" before introducing the abbreviation (CON) in the revised manuscript.

- L88, 89: I suggest using p < 0.01

We have revised the manuscript accordingly and applied this change for improved clarity and consistency of reporting.

- L99: I suppose that “short-term increase in HR” should not be stated in the conclusion because it is not a performance improvement (we focus only on important results about the subject: impact of AI on performance during TSD at least in the conclusion)

Thank you very much for your suggestion regarding the "short-term increase in HR" in our conclusion. We understand your point about focusing on key performance outcomes related to the impact of AI during TSD. However, we have decided to retain this information in our conclusion due to its importance in drawing parallels to previous studies on the same topic. We believe mentioning all relevant variables in the conclusion will ensure comprehensiveness and transparency allowing for a holistic understanding of the results.

- L101: I suggest that “the no changes” should not be stated in the conclusion.

Thank you for your suggestion regarding the reporting of unchanged results in the conclusion. While we appreciate your input, we believe it's essential to maintain transparency by presenting all our findings, regardless of whether they show change or not. Our intention is to provide a complete view of our experiment, which includes the unchanged variables. We, therefore, decided not to remove this information from the conclusion.

- L101, 102: The conclusion should be tempered and corrected; “AI could considerably improve performance in similar conditions and tests regardless of TSD (and not any duration of SD)”

Thank you for your valuable suggestion on tempering our conclusion. In line with your feedback, we have revised the text accordingly. It now reads: "These results suggest that AI could potentially be useful in some military scenarios, regardless of sleep deprivation." This new phrasing more accurately reflects our findings.

- L103: keywords are not well selected; I suggest adding Performance (not tests), Ammonia Inhalants (not Ergogenic Aids), Soldiers or Army (not both), and Sleep Deprivation (not sleep loss).

Thank you for your suggestions regarding the selection of keywords. We have adopted your recommendations of replacing "tests" with "Performance" and keeping "Army" while removing "Soldiers". However, we have chosen to retain "Ergogenic Aids" and "sleep loss" as they are broader terms that encapsulate "Ammonia Inhalants" and "Sleep Deprivation", respectively, which are already included in main tittle of manuscript. These keywords also align with the title of our manuscript and we believe it will aid in its visibility and reach.

Introduction

- L104: I suppose that there is a lack of general introduction concerning the disrupted sleep-wake cycle in everyday life (stress, shift work, jetlag…) before moving on to the part that specifies its relation with the military.

Thank you for your comment to provide a more general introduction on the disrupted sleep-wake cycle. To address this, we have included a new section of following text in our introduction: 

“Sleep is a fundamental yet often undervalued physiological process, indispensable for maintaining physical and mental well-being (1). It is an essential component of our circadian rhythms, which have been found to profoundly influence both cognitive and physical performance (2). However, these rhythms can easily be disrupted due to various factors such as stress, jetlag, travelling, or night-shift work. Such disruptions can result in cognitive and physical deficits, poorer mental health, and elevated health risks (3,4). Therefore, a deeper insight into the relationships between disruption to sleep and their combined effect on cognitive and physical performance is of paramount importance. This knowledge, while vital for all, may be especially critical for at-risk occupations such as military personnel. It has the potential to significantly influence their health and combat readiness (5). Given this, the optimization of health, well-being, and overall performance could be particularly crucial for them (6,7). The necessity of maintaining adequate sleep quality and quantity thus becomes evident, underscoring the integral role sleep plays in the well-being and performance of military personnel.”

- L105: I suggest adding another more general scientific definition of “sleep” before this specific definition.

In line with your feedback, we have revised the text as follows: "Sleep is a fundamental yet often undervalued physiological process, indispensable for maintaining physical and mental well-being (1). It is an essential component of our circadian rhythms, which have been found to profoundly influence both cognitive and physical performance (2). However, these rhythms can easily be disrupted due to various factors such as stress, jetlag, travelling, or night-shift work. Such disruptions can result in cognitive and physical deficits, poorer mental health, and elevated health risks (3,4).” We believe this change provides a clearer and more comprehensive definition of sleep, enhancing the understanding of our manuscript. Thank you for your constructive feedback.

L116, 117: The sentence should be reformulated; I suggest this correction; to various “physical”, cognitive, and behavioral impairments, such as fatigue and “decreased” attention and reaction time.

Thank you for your suggestion to reformulate our sentence on lines 116 and 117. We have added "physical" to the sentence as per your recommendation. However, regarding your suggestion to change "impaired" to "decreased", we have decided to retain the term "impaired". We believe that "impaired" offers a more comprehensive understanding, indicating not only a potential decrease in attention and reaction time, but also the possibility of increased errors.

- L136: I suggest using “psycho-stimulant” rather than “stimulant” (more accurate)

Thank you for your suggestion to use "psycho-stimulant" instead of "stimulant". However, to avoid potential confusion or debate over the distinction between these two terms, we have chosen to eliminate the word "stimulant" entirely from the revised manuscript.

-L139: I suppose that this study is very interesting about this topic and it could be added as a recent reference (Repeated low-dose caffeine ingestion during a night of total sleep deprivation improves endurance performance and cognitive function in young recreational runners: A randomized, double-blind, placebo-controlled study (2022)

https://doi.org/10.1080/07420528.2022.2097089

Thank you for suggesting the inclusion of this recent study. We agree that it's a valuable reference for our topic. We have now added the text "and exhaustive time on 400-meter run by 10%" to our manuscript, citing the recommended study.

-L143: “The potential positive effects of caffeine are likely to manifest about 1 hour after ingestion relative to many reviews and meta-analysis (and not between 10 and 60 min) (Reference: Grgic J, Grgic I, Pickering C, et al. Br J Sports Med 2020;54:681–688)

We have incorporated your comment into our revisited manuscript and added the following text: "Furthermore, multiple reviews and meta-analyses (28) show the potential positive effects of caffeine are likely to manifest about 1 hour after ingestion relative to, which may not be sufficient if immediate assistance is required.”

-L156: I suppose that the duration effect is also an important thing. Is there any information about how long it last for AI (the half-life of caffeine lasts between 2.5 and 4.5 h in humans)?

Thank you for your question about the duration effect of ammonia inhalants (AI). While it is indeed an important factor, to our best knowledge and as described in our previously published review (DOI: 10.1519/SSC.0000000000000630), no research has specifically reported the half-life of ammonia inhalants. However, it has been noted that AIs likely produce significant cerebrovascular, cardiovascular, and respiratory responses 7.5–15 seconds after inhalation (possibly up to under 30 seconds), but then these variables return to baseline values after 30 seconds. As such, it appears that the timing of inhalation before performance may be one of the most significant conditions when using AIs as an ergogenic aid.

-L163: I wonder if there are negative effects of AI in the literature (you should report in addition to the test results if there are any observed negative effects (during the experiments) of this stimulant in the discussion section).

We appreciate your attention to the potential negative effects of Ammonia Inhalants (AI). We have indeed considered this crucial aspect and included it in our discussion. To briefly summarize, exposure to large doses of concentrated ammonia vapors can result in injuries such as allergic reactions, respiratory airway burns, and distress, among others. Furthermore, although commercially available AI capsules release significantly lower amounts of ammonia vapors, some potential risks remain, including allergic reactions requiring medical attention, possible masking of head injury symptoms in contact sports, and potential respiratory difficulties in individuals with existing respiratory problems.

In our manuscript, we have expanded discussion part upon these points, using the following text: “Acknowledging the potential risks of AI, it is crucial to mention that it can be toxic under certain conditions, potentially cause allergic reactions, mask concussive symptoms in contact sports, and exacerbate breathing difficulties for those with respiratory issues. These risks emphasize that despite its legality in the U.S. for reviving fainting individuals, the use of AI for enhancing athletic performance should be carried out under caution and professional supervision (29). However, it is worth noting that we did not observe any adverse effects associated with its use during our research.”

-L164-168: This paragraph should clearly describe the objectives of the study: TSD (36 h), cognitive (SRT…) physical (JH) military personnel (18 male).

Thank you for your suggestion. We agree that explicitly stating the objectives could provide clarity. However, we believe that providing this information too early might potentially disrupt the flow of the introduction. As this information is comprehensively detailed in the methods section just a few paragraphs later, we have decided to maintain the current structure to ensure a smooth reading experience. We appreciate your valuable insight and will consider this feedback for future manuscripts.

Methods

-L171: The VO2max is also an important measurement (if it was done, add it)

Thank you for your suggestion. While VO2max is indeed an important measure, in this kind of studies, our primary focus was on cognitive and short-term physical variables. Given this focus, we did not consider VO2max a necessary measure for our research purposes. We appreciate your input and will certainly consider such aspects for future studies where long-term endurance might be more pertinent.

-L173: As outlined before, “cadets” need to be explained in parentheses (“young armed”, “police force”) to be clearer for readers.

To address this, we have revised the manuscript and now refer to the participants as "military cadets".

-L184: The participants' sleep quality (and quantity) is a very important measurement for such a study. For example, the sleep quality of athletes could be determined using the questionnaire of Horne and Östberg (1976). This latter could inform about the participants rising times, bedtimes, and sleep disorders.

Thank you for your insightful suggestion about monitoring participants' sleep quality and quantity. We actually collected such data using actigraphs and polysomnography throughout the study, and participants completed various sleepiness scales and sleep habit forms. However, we decided not to include these findings as we consider them beyond the scope of this particular paper. However, we are working on another, more comprehensive manuscript that includes these data on a larger scale. 

-L192: I suppose that the time should be written in a scientific article: 18:00 h and not in the military version: 1800 h (to be replaced throughout the manuscript)

Thank you for bringing this to our attention. We have amended the manuscript accordingly and replaced all instances of military time format with the standard scientific notation (i.e., 18:00 h).

-L196: Is there a familiarization (before the testing day) with the sleep deprivation (36 hours) to be familiar with the experimental process and the sleep adaptation? If yes, describe this familiarization.

In the manuscript, we have outlined the familiarization process stating, "They then completed a series of questionnaires addressing psychological and physiological health, which were followed by a general familiarization of the layout of the facility (i.e., location of the bathrooms, testing stations, etc.). During this familiarization, the participants were also familiarized with the specific testing procedures and practiced each of the required tasks."

Furthermore, to emphasize the participants' prior experience with sleep deprivation, we have added the following statement in the Participants section: "Additionally, the participants selected for this study were well-acquainted with total sleep deprivation, having experienced it during their military duties previously." We believe these statements adequately address the familiarization with both the experimental process and sleep deprivation.

-L202: Following “from”, we should put “to”.

We have made the necessary corrections in our manuscript and replaced "from" with "to" where appropriate.

-L202: How to be sure that participants have really passed 8 hours of recovery sleep? In the same context; how to be sure that the circadian rest-activity rhythms of the 18 subjects included in the study are well synchronized? It is obvious that only the use of devices for recording activity (actimeter, polysomnograph) or temperature, which are considered marker rhythms, could confirm the synchronization. Unfortunately, this was not carried out in this study.

Thank you for this comment. We appreciate your concern about the confirmation of sleep duration and synchronization of circadian rhythms. Indeed, to ensure the accuracy of our study, we had the participants wear actigraphs throughout the duration of the experiment, and they also underwent polysomnography during their recovery sleep. Polysomnography EEG activities were scored according to international criteria [Iber, C. (2007). The AASM manual for the scoring of sleep and associated events: rules, terminology, and technical specification] by trained evaluators. On average, the participants achieved 8 hours and 28 minutes of recovery sleep (+/- 11 minutes) from polysomnography testing and 7 hours and 45 minutes (+/- 30 minutes) from actigraphs testing from Saturday to Sunday.

However, for ease of reporting and interpretation, we approximated this time to 8 hours in the manuscript. We understand your interest in the detailed data of sleep duration and quality, as well as circadian rhythms. However, these data are being used in a separate, larger-scale research paper that we are currently preparing for submission. We believe that this approach prevents the duplication of results across our publications. We hope this clarifies our decision for the methodology of this study.

-L210: I suggest correcting as follows: “either with ammonia inhalants (AI) or with control (CON)”

Thank you for your suggestion on revising the sentence structure. We value your input as we strive to make our manuscript as clear and reader-friendly as possible. However, for the sake of consistency throughout the text, we have decided to retain the current format. We believe this approach maintains the flow of the document and minimizes potential confusion. We hope you understand and thank you again for your valuable feedback.

-L213-217: The experimental protocol was not blinded in this study. This latter could influence the performance outcomes (participants could be more motivated with AI supplement) which could limit the study conclusions. This should be added in the section of the study limitations.

We appreciate your observation on the lack of blinding in our study and the potential influence it could have on the performance outcomes. Indeed, the distinct smell and immediate physical response after inhalation of ammonia inhalants (AIs) render a true placebo control virtually impossible. In response to your suggestion, we have added the following to the limitations section of our manuscript: "Additionally, one limitation of this study was the absence of a placebo condition due to the distinct characteristics of AIs. Participants can likely distinguish AIs from any potential placebo due to their strong smell and immediate physical response after inhalation. Though some placebo substances like menthol oil or Vicks VapoRub have been used in other studies, these also have identifiable smells that can potentially compromise the blinding process. Consequently, participants in our study likely knew when they were not inhaling an AI. Nonetheless, no studies have reported an increase in performance superior to AI inhalation for placebo or control conditions, indicating a limited placebo effect in this context." We believe this paragraph effectively addresses the limitation you highlighted, and we appreciate your thoughtful suggestion.

-L224: I wonder if the mean energy intake per capita/day was calculated for participants. If yes, please add it.

Yes, we indeed calculated the mean energy intake per day for each participant. However, as part of a larger research project, we are in the process of preparing another manuscript that focuses on the effects of sleep deprivation on hunger levels, among other variables. To avoid redundancy and possible data saturation, we decided not to include this specific piece of information in the current study. We believe that including the data on energy intake here could potentially overshadow the main findings of our study. However, we assure you that we have carefully monitored and accounted for this variable during our experiment. We hope this clarification meets your understanding.

-L252: It is useless to rewrite “in a randomized order” because it was already stated in the methods section. (To be corrected throughout this section)

Thank you for your observation regarding the repetition. However, we have chosen to retain this phrase in certain sections to continually remind readers about the randomization of the procedure, which is a key aspect of our study design. This choice was made to reinforce the study's methodological rigor and prevent potential confusion. We hope this explanation helps, and we value your thoughtful feedback.

-L258: I suggest starting by giving the quality measured by the test (accuracy, speed, strength) and the type of test (cognitive, physical, psychological), before the description, and this for all 5 tests (for more clarity).

Thank you for your suggestion to start with the quality measured by each test (accuracy, speed, strength) and the type of test (cognitive, physical, psychological) prior to describing the tests in detail. We appreciate your attention to the clarity of our methodology. However, we have chosen to describe the tests chronologically, consistent with the figure that outlines the order of the tests. We believe that this approach will help readers understand the sequence of events in our study and prevent any potential confusion. We hope this clarifies our decision, and we appreciate your thoughtful feedback.

-L319-323: The familiarization information should be written in the methods section. In this section, we only describe the realization of the test.

Thank you for your suggestion to relocate the familiarization information to the methods section. However, we believe it's important to keep this information in the current section. The test in question is a military-specific test with a particular rifle, which is not commonly done. Therefore, we wanted to emphasize the reliability of this unique test in its immediate context. The familiarization information demonstrates that participants were adequately prepared for this specialized procedure, which adds to the reliability of the results. We appreciate your understanding in this matter and value your thoughtful feedback.

-L342: All abbreviations should be described; (LL, UL)

We have updated the text to include the explanation for "LL" and "UL". The revised text now reads: "Cohen’s d with 95% lower limit (LL) and upper limit (UL) confidence intervals [LL, UL]." Your attention to detail is greatly appreciated and contributes to the overall clarity of the manuscript.

Results

-L364: I suggest replacing “incorrect trials” with “incorrect detections” for more clarity.

Thank you for your suggestion to replace "incorrect trials" with "incorrect detections" for greater clarity. However, we respectfully disagree with this recommendation in the context of our study. In the case of simple reaction time tasks, the term "trial" refers to each individual attempt or response made by the participant. Therefore, "incorrect trials" accurately reflects the number of times the participants made incorrect responses during the task.

On the other hand, the term "detections" typically implies the identification or recognition of specific stimuli or targets. This terminology is more commonly used in tasks such as signal detection or target identification. In our study, the focus was on the accuracy of the participants' overall responses, rather than specifically detecting or recognizing stimuli.

Considering these factors, we believe that using "incorrect trials" is appropriate and clearer in the context of our study. We appreciate your suggestion and value your input in improving the clarity of our manuscript.

-L379: I also suggest replacing “condition” with “ammonia”, which would be clearer and easier to follow for readers.

Thank you for your suggestion to replace "condition" with "ammonia" for greater clarity. While we appreciate your input, we respectfully disagree with this recommendation.

In our study, the term "condition" refers to the experimental conditions being compared, specifically the ammonia inhalant condition and the control condition. Using the term "ammonia" instead of "condition" may potentially create confusion for readers who are unfamiliar with the study design or the specific terminology used in the field.

By maintaining the term "condition," we ensure consistency and clarity throughout the manuscript, allowing readers to easily follow and understand the experimental setup. We value your feedback and always strive to enhance the clarity of our work.

-L390-443: There is no need to present all the results especially when they are unnecessary (or not important for the conclusions) and will bother readers.

Thank you for your valuable comment. Regarding your suggestion about the data presented on lines 390-443, we understand your perspective about not presenting all results, particularly those deemed less critical to the main conclusions. Your concern about the potential burden on the readers is duly noted. However, our intention behind presenting these results in detail was driven by a strong commitment to transparency and consistency. Our aim was to provide a comprehensive overview of all the data and outcomes we obtained during our study. We believe that every reader, with varying interests and focus points, might find different parts of the results relevant. Thus, providing a complete set of results could serve to meet the diverse needs of our audience. We understand the importance of maintaining a balance between detail and brevity to not overburden the reader. In response to your comment, we will undertake a careful review of the specified section to ensure it is as concise and reader-friendly as possible while still maintaining the comprehensive presentation of our results.

Discussion

- I suggest that this section should be rewritten in a more precise way. It should focus on essential results about the main subject: the impact of TSD on the tests selected, and the impact of AI on performance tests (by describing the scientific reasons about previous similar works, without repeating results or useless information).

We appreciate your valuable feedback on improving the precision and focus of our discussion. Following your suggestion, we have thoroughly reviewed and rewritten the specified section to highlight the essential results regarding the impact of total sleep deprivation (TSD) and ammonia inhalants (AI) on performance tests. Unnecessary information, such as specific statistical details like 'condition × time interactions', has been removed to streamline the text and increase readability. In addition, to enhance consistency and navigability, we've added relevant subheadings - "Effect of Total Sleep Deprivation" and "Effects of Ammonia Inhalants".

We believe these revisions address your concerns effectively, ensuring a focused and concise discussion section. We are grateful for your guidance and continued assistance in refining our manuscript.

-L516: This reference is important and could also be added; Amir Khcharem, Makram Souissi, Rim Atheymen, Lobna Ben Mahmoud & Zouheir Sahnoun (2020): Effects of caffeine ingestion on 8-km run performance and cognitive function after 26 hours of sleep deprivation, Biological Rhythm Research, DOI: 10.1080/09291016.2020.1778975

Thank you for your suggestion to include the reference by Khcharem et al., 2020, in our manuscript. We appreciate your efforts in identifying this relevant piece of work which aligns with the context discussed on line 516. We have incorporated this reference into the stated section. 

Limitations

-L705: Some limitations of the study were not stated although they could influence the conclusions; the sample size is quite small; 18 and only men were used), the experimental protocol was not blinded, and qualitative and quantitative evaluation of sleep were not carried out.

Thank you for highlighting the limitations that might influence the conclusions of our study. Your input regarding the sample size, absence of blinding, and the lack of qualitative and quantitative evaluation of sleep is very much appreciated.

In response to your comments, we have incorporated additional text in the manuscript. For the sample size, we agree that a larger and more diverse cohort would provide a more comprehensive view. However, we had 18 participants for logistical and resource reasons, which we believe is adequate as substantiated by the sensitivity analysis described in our statistics section. As for the lack of blinding, we acknowledge that blinding is a crucial element in many experimental designs. However, due to the nature of the study and the use of ammonia inhalants, blinding was not feasible. We have added a text in the limitations section to further elaborate on this point. While we do possess qualitative and quantitative sleep evaluation data, we chose not to include them in this manuscript. Our decision was guided by considerations for readability and consistency. Furthermore, we're currently preparing a more extensive manuscript where these data, including findings from actigraphy, will be thoroughly presented and discussed. Including them here might lead to instances of double-publishing, which we aim to avoid. This strategic limitation is included in our manuscript (see lines 746 to 770 in the revised manuscript). We sincerely appreciate your thoughtful suggestions and believe that the clarifications added in response to your points will provide a more accurate and comprehensive perspective for our readers.

Conclusion

I suggest that the conclusion needs to be reformulated; there is no need to repeat the results in this section and with the same abbreviations, (e.g. ammonia inhalant markedly improved the reaction time, jumping, and sensation of pain of soldiers). It should conclude the main results in a good manner. In addition, it should be tempered (e.g. in similar testing conditions of prolonged sleep deprivation). It should also finish with a recommendation for future studies about this topic (e.g., Further investigations including females and using different tests are needed…)

We sincerely thank you for your constructive feedback on our conclusion section. Your suggestion to refrain from repeating the results verbatim, temper the interpretation, and include recommendations for future research was truly insightful.

In accordance with your recommendations, we have restructured our conclusion. We've incorporated a tempered discussion of our main findings while emphasizing the study's specific context, and concluded with recommendations for future investigations.

The revised conclusion now reads: "These results indicate potential applications of AI in specific military scenarios, regardless of the presence or the length of total sleep deprivation. Nevertheless, caution is warranted in generalizing these results, given the study's limitations, such as the small, all-male sample size. Future research in this field would benefit from including more diverse participant groups and employing a wider range of military-related performance tests. This would provide a more comprehensive understanding of AI's effects under conditions of sleep deprivation, informing its potential utility in various operational contexts." We believe these amendments respond effectively to your concerns and provide a more rounded conclusion. Your astute observations and guidance have significantly contributed to improving the quality of our manuscript.

-L724: I suppose that the sentence “No matter the length of sleep deprivation” was inaccurate and should be replaced.

Thank you for pointing out the potential inaccuracy in the sentence at line 724. We agree that the phrase "No matter the length of sleep deprivation" could be misinterpreted.

In response to your suggestion, we have revised this statement to provide a clearer, more precise assertion. The updated sentence now reads: "regardless of the presence or the length of total sleep deprivation." We appreciate your diligent attention to detail and your constructive feedback, which have been instrumental in improving the clarity and accuracy of our manuscript.

Reviewer #2: A single cohort study of 18 healthy male cadets investigated the acute effects of ammonia inhalants (AI) versus control on cognitive and physical performance during a 36 hour period of total sleep deprivation (TSD). ANOVA methods were used to analyze the main objectives of the trial. The conclusions are unclear.

Major revision:

If the interaction effect is significant, provide an interpretation of the results, but do not test main effects because the tests for main effects are uninteresting in light of significant interactions. If interaction effects are non-significant, drop the interaction effects from the model and test the main effects. Determining which results to present when testing interactions is often a multi-step process. Typically these type of results are presented in table format because they are more easily comprehendable.

Thank you for taking the time to review our manuscript and for your insightful comments. Your feedback is instrumental in refining the quality of our work, and we greatly appreciate your suggestions on how to clarify the presentation and interpretation of our statistical analyses.

Addressing your first point regarding the unclear conclusions, we have made substantial revisions to the discussion section to clearly elucidate our conclusions based on the data and statistical analyses. We believe that the revised manuscript now provides a coherent understanding of the study’s outcomes.

Regarding your recommendation on how to handle and present interaction effects and main effects in the ANOVA analyses, we acknowledge the importance of correctly interpreting and reporting these effects. Following your guidance, we have adopted the multi-step process for presenting results when testing interactions:

Where the interaction effect was significant, we have provided an in-depth interpretation of these results, as you suggested. We refrained from testing or reporting the main effects in these cases, as per your advice.

In cases where the interaction effects were non-significant, we have removed them from the model and focused on testing the main effects as recommended.

The text referring to non-significant interactions has been deleted in both the results and abstract sections, as you suggested.

We hope that these revisions address your concerns adequately. We are grateful for your expert review, as it has significantly contributed to improving the rigor and clarity of our manuscript.

Journal Requirements:

2 We note that the grant information you provided in the ‘Funding Information’ and ‘Financial Disclosure’ sections do not match.

3. Please include your tables as part of your main manuscript and remove the individual files. Please note that supplementary tables (should remain/ be uploaded) as separate "supporting information" files.

4. We note that the original protocol that you have uploaded as a Supporting Information file contains an institutional logo. As this logo is likely copyrighted, we ask that you please remove it from this file and upload an updated version upon resubmission.

Thank you for outlining the additional requirements necessary for the revision of our manuscript. We appreciate your guidance on adhering to PLOS ONE's specific standards.

1. Style Requirements: We have thoroughly revised the manuscript to align with PLOS ONE’s style guidelines, utilizing the provided templates for both the main body and title, authors, and affiliations.

2. Funding Information: We apologize for the discrepancy in the grant information. This has been corrected to accurately reflect the grant numbers in the 'Funding Information' section.

3. Tables: As advised, we have incorporated the table into the main manuscript and removed the individual files.

4. Protocol File: We have removed the institutional logo from the original protocol in the Supporting Information file due to copyright considerations and uploaded an updated version.

We trust that these changes fulfill the journal's requirements, and we look forward to your response.

---

## [Decision Letter · Decision Letter 1]

18 Sep 2023

PONE-D-23-07437R1Effects of 36 hours of sleep deprivation on military-related tasks: can ammonium inhalants maintain performance?PLOS ONE

Dear Dr. Malecek,

Thank you for submitting your manuscript to PLOS ONE. After careful consideration, we feel that it has merit but does not fully meet PLOS ONE’s publication criteria as it currently stands. Therefore, we invite you to submit a revised version of the manuscript that addresses the points raised during the review process.

Please see a few minor comments from reviewer 2 below, which we now invite you to address.

We look forward to receiving your revised manuscript.

Kind regards,

Hanna Landenmark

Staff Editor

PLOS ONE

Journal Requirements:

Reviewers' comments:

Reviewer's Responses to Questions

**Comments to the Author**

1. If the authors have adequately addressed your comments raised in a previous round of review and you feel that this manuscript is now acceptable for publication, you may indicate that here to bypass the “Comments to the Author” section, enter your conflict of interest statement in the “Confidential to Editor” section, and submit your "Accept" recommendation.

Reviewer #1: All comments have been addressed

Reviewer #2: (No Response)

2. Is the manuscript technically sound, and do the data support the conclusions?

Reviewer #1: Yes

Reviewer #2: Yes

3. Has the statistical analysis been performed appropriately and rigorously? 

Reviewer #1: Yes

Reviewer #2: Yes

4. Have the authors made all data underlying the findings in their manuscript fully available?

Reviewer #1: Yes

Reviewer #2: Yes

5. Is the manuscript presented in an intelligible fashion and written in standard English?

Reviewer #1: Yes

Reviewer #2: Yes

6. Review Comments to the Author

Reviewer #1: The revision of the manuscript is satisfactory and the authors have addressed all the questions raised by the reviewers.

Reviewer #2: Minor revisions:

1- Line 185: Clarify that the provided summary measures represent age, height, and weight.

2- Line 344: Typographical error: Drop the second "W" from "Shapiro-Wilk W test."

3- Line 367: Use a lower case t in t-test.

4- Results: For clarity, state that non-significant interaction effects were dropped from the models before testing main effects. Tables of these results would help readers better grasp the them.

5- Define pbonf at its first appearance.

7. PLOS authors have the option to publish the peer review history of their article (what does this mean?). If published, this will include your full peer review and any attached files.

Reviewer #1: **Yes: **Amir Khcharem

Reviewer #2: No

---

## [Author Response · Author response to Decision Letter 1]

19 Sep 2023

Dear Editor,

We are deeply appreciative of the insightful comments provided on our manuscript. The reviewers' constructive feedback was instrumental, guiding us in refining and enhancing our work. After incorporating these suggestions, we believe the manuscript has become significantly stronger, more comprehensive, and better positioned to contribute meaningfully to the field.

Review Comments to the Author:

Reviewer #1: The revision of the manuscript is satisfactory and the

authors have addressed all the questions raised by the reviewers.

Thank you very much.

Reviewer #2: Minor revisions:

Thanks for your insightful feedback on our paper. We recognize the significance of your points and we incorporated them in our revised version to improve its rigor.

1- Line 185: Clarify that the provided summary measures represent age,

height, and weight.

Thank you for pointing out the need for clarification in line 185. We revised the sentence to explicitly indicate that the summary measures are for age, height, and weight. The revised sentence is: “Eighteen healthy male military cadets (age = 24.1 ± 3.0 years, height = 181.5 ± 6.3 cm, weight = 79.3 ± 8.3 kg, 4.0 ± 0.9 total years of service, all measurements reported as mean ± SD) serving at the Military department of Charles University participated in this study.”

2- Line 344: Typographical error: Drop the second "W" from "Shapiro-Wilk

W test."

Thank you for spotting the typographical error. We corrected it to "Shapiro-Wilk test" in the revised manuscript.

3- Line 367: Use a lower case t in t-test.

Thank you for your suggestion. We changed "T-test" to "t-test" in the revised manuscript.

4- Results: For clarity, state that non-significant interaction effects were dropped from the models before testing main effects. Tables of these results would help readers better grasp the them.

Thank you for your suggestion to clarify the handling of non-significant interaction effects in our analysis. We have added the sentence "Non-significant interaction effects were excluded from the models prior to the examination of main effects." to the Results section (at line 350), to address this point. We believe this addition will enhance the clarity of our methodology.

Thank you also for suggesting the inclusion of tables to enhance readers' understanding of the results. While we acknowledge the value this would add, we have chosen to include all detailed tables in the supplementary materials available online. This approach allows interested readers to access all the specifics while maintaining the conciseness of the manuscript.

5- Define pbonf at its first appearance.

Thank you for pointing out the need to define “pbonf” at its first appearance. We have revised the text to clarify this term. The first sentence (at line 346) now reads: “When the ANOVA tests demonstrated a statistically significant condition × time (× time spans or × percentage difference) interaction or a statistically significant main effect for condition, time, time spans, or percentage difference, post-hoc comparisons of the mean differences were performed using the Bonferroni correction (pbonf).”

---

## [Decision Letter · Decision Letter 2]

20 Oct 2023

Effects of 36 hours of sleep deprivation on military-related tasks: can ammonium inhalants maintain performance?

PONE-D-23-07437R2

Dear Dr. Malecek,

We’re pleased to inform you that your manuscript has been judged scientifically suitable for publication and will be formally accepted for publication once it meets all outstanding technical requirements.

Kind regards,

Steve Zimmerman, PhD

Senior Editor, PLOS ONE

Additional Editor Comments (optional):

Reviewers' comments:

Reviewer's Responses to Questions

**Comments to the Author**

1. If the authors have adequately addressed your comments raised in a previous round of review and you feel that this manuscript is now acceptable for publication, you may indicate that here to bypass the “Comments to the Author” section, enter your conflict of interest statement in the “Confidential to Editor” section, and submit your "Accept" recommendation.

Reviewer #2: All comments have been addressed

2. Is the manuscript technically sound, and do the data support the conclusions?

Reviewer #2: (No Response)

3. Has the statistical analysis been performed appropriately and rigorously? 

Reviewer #2: (No Response)

4. Have the authors made all data underlying the findings in their manuscript fully available?

Reviewer #2: (No Response)

5. Is the manuscript presented in an intelligible fashion and written in standard English?

Reviewer #2: (No Response)

6. Review Comments to the Author

Reviewer #2: (No Response)

7. PLOS authors have the option to publish the peer review history of their article (what does this mean?). If published, this will include your full peer review and any attached files.

Reviewer #2: No

---

## [Editor Report · Acceptance letter]

6 Nov 2023

PONE-D-23-07437R2 

Effects of 36 hours of sleep deprivation on military-related tasks: can ammonium inhalants maintain performance? 

Dear Dr. Malecek:

I'm pleased to inform you that your manuscript has been deemed suitable for publication in PLOS ONE. Congratulations! Your manuscript is now with our production department. 

Kind regards, 

on behalf of

Dr Steve Zimmerman 

Staff Editor

PLOS ONE